# Time-VLM: Exploring Multimodal Vision-Language Models for Augmented Time Series Forecasting

**Siru Zhong** [1]  **Weilin Ruan** [1]  **Ming Jin** [2]  **Huan Li** [3]  **Qingsong Wen** [4]  **Yuxuan Liang** [1]

## Abstract

Recent advancements in time series forecasting have explored augmenting models with text or vision modalities to improve accuracy. While text provides contextual understanding, it often lacks fine-grained temporal details. Conversely, vision captures intricate temporal patterns but lacks semantic context, limiting the complementary potential of these modalities. To address this, we propose Time-VLM, a novel multimodal framework that leverages pre-trained Vision-Language Models (VLMs) to bridge temporal, visual, and textual modalities for enhanced forecasting. Our framework comprises three key components: (1) a Retrieval-Augmented Learner, which extracts enriched temporal features through memory bank interactions; (2) a Vision-Augmented Learner, which encodes time series as informative images; and (3) a Text-Augmented Learner, which generates contextual textual descriptions. These components collaborate with frozen pre-trained VLMs to produce multimodal embeddings, which are then fused with temporal features for final prediction. Extensive experiments demonstrate that Time-VLM achieves superior performance, particularly in few-shot and zero-shot scenarios, thereby establishing a new direction for multimodal time series forecasting. Code is available at https://github.com/CityMind-Lab/ICML25-TimeVLM.

## 1. Introduction

Time series data captures temporal signal evolution and underpins forecasting in diverse domains such as finance

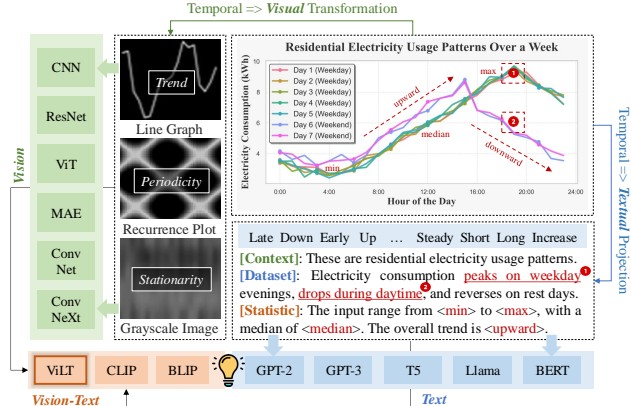

Figure 1: Our Time-VLM combines text (Right) and vision (Left) modalities to augment time series forecasting.

(Idrees et al., 2019), climate (Karevan & Suykens, 2020), energy (Deb et al., 2017), and transportation (Zheng & Huang, 2020). Accurate forecasting supports proactive risk mitigation, efficient resource allocation, and data-driven decision-making. Traditional models like ARIMA, while historically dominant, struggle to capture complex nonlinear patterns. In contrast, deep learning methods—from recurrent neural networks (RNNs) (Medsker et al., 2001) to Transformer-based architectures (Li et al., 2019; Wu et al., 2021; Zhou et al., 2021; Liu et al., 2022a; Zhou et al., 2022; Nie et al., 2023)—leverage innovations such as patch-based feature extraction, auto-correlation mechanisms, and frequency decomposition to model complex temporal dynamics. Despite their success, these models often fail to generalize across domains or adapt to data-limited scenarios, particularly few-shot and zero-shot settings (Liang et al., 2024).

To overcome these issues, researchers have turned to *augmenting time series forecasting with additional modalities*, such as *text* and *images*, which provide complementary information that can enhance predictive accuracy:

- **Text-Augmented Models:** Textual data offers valuable semantics crucial for accurate forecasting. For instance, contextual descriptions or dataset statistics can significantly enrich the understanding of time series patterns (Figure 1, right). Methods like Time-LLM (Jin et al., 2024) and UniTime (Liu et al., 2024b) leverage the superior pre-trained inference capabilities of Large Language

[1]The Hong Kong University of Science and Technology (Guangzhou), China [2]Griffith University, Australia [3]Zhejiang University, China [4]Squirrel Ai Learning, USA. Correspondence to: Yuxuan Liang <yuxliang@outlook.com>.

*Proceedings of the 42nd International Conference on Machine Learning*, Vancouver, Canada. PMLR 267, 2025. Copyright 2025 by the author(s).

Models (LLMs) by mapping time series into textual representations. However, these approaches encounter two key challenges: (1) the *modality gap* between continuous time series and discrete text leads to information loss during representation alignment, and (2) pre-trained language knowledge is rare for capturing fine-grained temporal patterns, limiting their ability to learn nuanced dynamics.

- **Vision-Augmented Models:** Visual representations of time series—such as line graphs, recurrence plots, or grayscale images—enable models to exploit spatial patterns embedded within temporal data. By transforming sequences into images, methods such as CNNs, ViTs, and MAEs can extract hierarchical features that reveal latent temporal relationships (Figure 1, left). Recent studies (Wu et al., 2023b; Wang et al., 2024; Chen et al., 2024) have demonstrated the natural alignment between time series and vision, as both are continuous and share structural similarities, allowing pre-trained vision models to effectively encode temporal hierarchies. However, these methods lack semantic interpretability, restricting their capacity to incorporate domain-specific knowledge.

Despite advancements in text- and vision-augmented models, integrating both modalities with time series remains underexplored. Current approaches often *focus on single modalities, failing to harness their combined strengths*. To address this gap, we propose Time-VLM, a novel framework that leverages pre-trained VLMs to enhance time series forecasting by unifying temporal, visual, and textual information. VLMs provide a promising foundation, as they excel at aligning visual and textual modalities, making them well-suited for incorporating temporal information to unify the three modalities. By projecting time series into a unified vision-language semantic space, Time-VLM enables rich cross-modal interactions, combining the strengths of both modalities while mitigating their individual limitations. In this paradigm, each modality contributes uniquely: text provides semantic context, vision captures spatial-temporal patterns, and time series encodes sequential dynamics.

Specifically, Time-VLM introduces three key components: (1) a *Retrieval-Augmented Learner* that processes raw time series data through patch-based feature extraction and memory bank interactions to generate enriched temporal representations, capturing both local and global dependencies; (2) a *Vision-Augmented Learner* that adaptively transforms time series into images using multi-scale convolution, frequency encoding, and periodic encoding, preserving both fine-grained details and high-level structures; and (3) a *Text-Augmented Learner* that generates rich textual context (e.g., statistics and dataset descriptions) to complement the visual representations. These modules collaborate with VLMs to integrate temporal, visual, and textual modalities, producing accurate forecasts through a fine-tuned predictor.

Our key contributions can be summarized as follows:

- We propose the first framework that unifies temporal, visual, and textual modalities by leveraging their complementary strengths for enhanced time series forecasting.

- We introduce a retrieval-augmented learner for hierarchical temporal feature enhancement, a vision-augmented learner for adaptive time-series-to-image transformation, and a text-augmented learner for contextual prompt generation, enabling seamless integration with VLMs.

- Extensive evaluations show Time-VLM's strong performance, especially under data-scarce conditions, offering a brand new paradigm for multimodal time series research.

## 2. Related Work

**Text-Augmented Models for Time Series Forecasting.**

The success of LLMs has inspired their application to time series forecasting. Methods like LLMTime (Gruver et al., 2023) and LLM4TS (Chang et al., 2023) tokenize time series for autoregressive prediction but inherit limitations such as poor arithmetic and recursive reasoning. Approaches including GPT4TS (Zhou et al., 2023) and TimeLLM (Jin et al., 2024) project time series into textual representations to leverage LLMs' reasoning capabilities, yet face challenges like the modality gap and limited temporal adaptability of word embeddings. UniTime (Liu et al., 2024b) and TimeFFM (Liu et al., 2024a) incorporate domain knowledge and federated learning, respectively, but remain constrained by their exclusive dependence on textual modeling.

**Vision-Augmented Models for Time Series Forecasting.**

Vision provides a natural way to preserve temporal patterns in time series data. Early approaches apply CNNs to matrix-formed time series (Li et al., 2020; Sood et al., 2021), while TimesNet (Wu et al., 2023b) introduces multi-periodic decomposition for unified 2D modeling. VisionTS (Chen et al., 2024) pioneers the use of pre-trained visual encoders with grayscale time series images, and TimeMixer++ (Wang et al., 2024) advances the field through multi-scale frequency-based time-image transformations. Despite their effectiveness in temporal modeling, these methods often lack semantic context, limiting their ability to utilize high-level contextual information for prediction.

**Vision-Language Models.**

VLMs like ViLT (Kim et al., 2021), CLIP (Radford et al., 2021), and ALIGN (Jia et al., 2021), have transformed multimodal understanding by aligning image and text representations. Recent progress, like BLIP-2 (Li et al., 2023) and LLaVA (Liu et al., 2023), further enhance multimodal reasoning. However, VLMs remain underexplored for time series analysis. Our work bridges this gap by leveraging VLMs to integrate temporal, visual, and textual modalities, addressing the limitations of unimodal approaches.

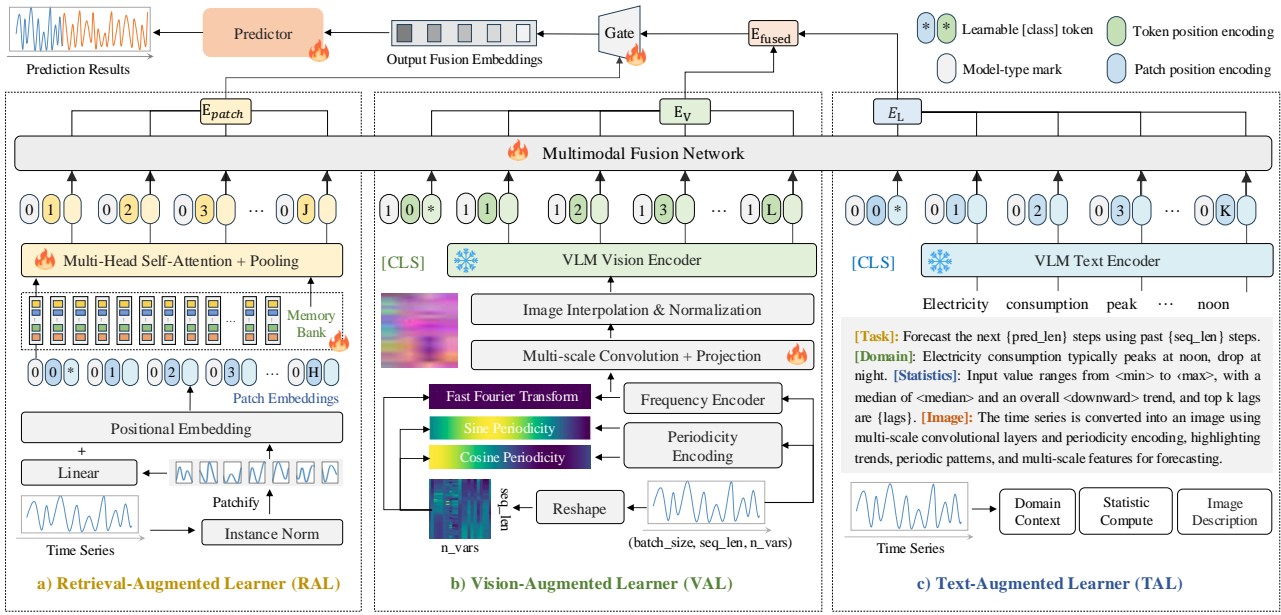

Figure 2: Overview of the Time-VLM framework.

## 3. Methodology

To address the limitations of single-modality approaches and leverage the complementary strengths of visual, textual, and temporal modalities, we propose Time-VLM, a unified framework that integrates these modalities for enhanced time series forecasting. As illustrated in Figure 2, the framework comprises three core components:

- **Retrieval-Augmented Learner (RAL)**: Extracts temporal features from raw time series patches and maintains a memory bank to refine patch embeddings through multi-head self-attention and pooling mechanisms, thereby preserving rich temporal representations and enhancing long-term dependency modeling for robust forecasting.

- **Vision-Augmented Learner (VAL)**: Transforms time series into informative three-channel images through multi-scale convolutions, frequency and periodic encoding. The images are processed by a frozen VLM vision encoder to extract hierarchical visual features, capturing both fine-grained details and high-level temporal patterns.

- **Text-Augmented Learner (TAL)**: Generates contextual textual prompts for input time series, including statistical features (e.g., mean, variance, trends), domain-specific context (e.g., electricity consumption patterns), and image descriptions. These prompts are encoded by a frozen VLM text encoder to produce textual embeddings.

The image and text embeddings, extracted by the VLM, are integrated with temporal memory features via a gated fusion mechanism, effectively capturing complementary information to improve forecasting accuracy. These enriched multimodal features are then processed by a fine-tuned predictor to generate precise and reliable forecasts.

### 3.1. Retrieval-Augmented Learner (RAL)

The RAL module extracts high-level temporal features via patch-based processing and retrieval-augmented memory mechanisms. It dynamically retrieves historical patterns and integrates them with current observations, adapting to complex time series structures. It operates in two key stages.

**Patch Embedding:** The input time series $x_{\text{enc}} \in \mathbb{R}^{B \times L \times D}$ is divided into overlapping patches of length $pl$ with stride $st$, where $B$, $L$, and $D$ denote the batch size, sequence length, and number of variables, respectively. Each patch is linearly projected into a $d_{\text{model}}$-dimensional latent space, and positional embeddings are added to preserve temporal order. This yields patch embeddings $E_{\text{p}} \in \mathbb{R}^{B \times N_p \times d_{\text{model}}}$, where $N_p = \frac{L-pl}{st} + 1$ represents the total number of patches.

**Retrieval-Augmented Memory:** A memory bank with maximum capacity $M$ stores historical patch representations $\mathcal{M} \in \mathbb{R}^{M \times d_{\text{model}}}$. During each forward pass, the current patch embeddings are averaged across the temporal dimension and added to the memory bank using a circular buffer update strategy, ensuring that the most recent patterns are retained. To better capture temporal dynamics, we introduce a hierarchical memory structure:

- **Local Memory:** Given current patch embeddings $P \in \mathbb{R}^{B \times N_p \times d_{\text{model}}}$, we retrieve top-$k$ similar patches from the memory bank $\mathcal{M}$ based on cosine similarity:

$$\text{sim}(P, \mathcal{M}) = P \cdot \mathcal{M}^\top, \qquad (1)$$

where $\mathcal{M} \in \mathbb{R}^{M \times d_{\text{model}}}$ stores historical patch representations. The retrieved patches are processed through a

two-layer MLP to extract local memory features:

$$M_{\text{local}}^{(i)} = \text{MLP}(\text{topk}(E_p^{(i)})), \quad i = 1, \ldots, B. \quad (2)$$

These features are averaged across patch dimension and combined with the original $P$ via a residual connection.

- **Global Memory:** To capture long-range dependencies, we apply multi-head self-attention over the current patch embeddings $P$, yielding contextualized representations:

$$\text{Attn}(P) = \text{MultiHead}(Q, K, V), \quad (3)$$

where $Q, K, V$ are linear projections of $P$. The global memory is obtained by temporal averaging:

$$M_{\text{global}} = \frac{1}{N_p} \sum_{i=1}^{N_p} \text{Attn}(P)_i. \quad (4)$$

The two memories are fused via a gated mechanism:

$$M_{\text{fused}} = \alpha \cdot M_{\text{local}} + (1 - \alpha) \cdot M_{\text{global}}, \quad (5)$$

where $\alpha$ is a learnable gating parameter that adaptively balances the contribution of local and global memory.

The output $M_{\text{fused}} \in \mathbb{R}^{B \times N_p \times d_{\text{model}}}$ captures high-level temporal patterns. This fused representation supports dynamic retrieval and integration with other modalities (e.g., vision or text), enabling adaptive context-aware forecasting.

### 3.2. Vision-Augmented Learner (VAL)

The VAL module adaptively transforms the input time series $x_{\text{enc}} \in \mathbb{R}^{B \times L \times D}$ into image representations, enabling fine-grained and high-level temporal pattern extraction via the VLM vision encoder. The process is in three steps:

**Frequency and Periodicity Encoding:** To capture spectral and temporal dependencies, the VAL module applies two complementary encoding techniques to the input time series, explicitly adding frequency and time-domain information.

1. **Frequency Encoding:** A Fast Fourier Transform (FFT) extracts frequency components from raw input $x_{\text{enc}}$ as:

$$\text{FFT}(x_{\text{enc}}) = \sum_{t=0}^{L-1} x_{\text{enc}}(t) \cdot e^{-2\pi i k t / L}, \quad (6)$$

where $k$ is the frequency index. The resulting frequency features are concatenated with the input time series, resulting in a tensor of shape $\mathbb{R}^{B \times L \times D \times 2}$.

2. **Periodicity Encoding:** Temporal dependencies are encoded using sine and cosine functions for each time step:

$$\text{encoding}(t) = \left[ \sin\left(\frac{2\pi t}{P}\right), \cos\left(\frac{2\pi t}{P}\right) \right], \quad (7)$$

where $P$ is the periodicity hyperparameter. These encodings are concatenated with the input time series, resulting in a tensor of shape $\mathbb{R}^{B \times L \times D \times 3}$. Complete periodic parameter settings can be found in Appendix A.

**Multi-scale Convolution:** The concat tensor is processed through multiple convolutional layers to extract hierarchical temporal patterns. A 1D convolutional layer captures local dependencies, transforming the input into $\mathbb{R}^{B \times D \times H_{\text{hidden}} \times L}$, where $H_{\text{hidden}}$ is the hidden dimension. Averaging along $D$ yields $\mathbb{R}^{B \times H_{\text{hidden}} \times L}$. Two 2D convolutional layers follow: the first halves the channel dimension, and the second maps features to $C$ output channels, producing the final output capturing both local and global temporal structures.

**Image Interpolation & Normalization:** The output tensor is resized to the desired image dimensions $(H, W)$ using bilinear interpolation. For a target pixel $(x, y)$, the interpolated value $\mathbf{I}(x, y)$ is computed as follows:

$$\mathbf{I}(x, y) = \sum_{i=1}^{2} \sum_{j=1}^{2} \mathbf{I}(x_i, y_j) \cdot w_{ij}, \quad (8)$$

$$\text{Inorm} = 255 \cdot \frac{\mathbf{I}_{\text{raw}} - \text{Min}(\mathbf{I}_{\text{raw}})}{\text{Max}(\mathbf{I}_{\text{raw}}) - \text{Min}(\mathbf{I}_{\text{raw}}) + \epsilon}, \quad (9)$$

where $(x_i, y_j)$ are the coordinates of the four nearest neighbors, $w_{ij}$ are weights based on relative distances, and $\epsilon = 10^{-5}$ prevents division by zero. Pixel values are scaled to $[0, 255]$ via min-max normalization, producing the normalized image $\mathbf{I}_{\text{norm}} \in \mathbb{R}^{B \times C \times H \times W}$ ($C$ is the number of channels). This ensures alignment with the VLM vision encoder's input distribution for effective feature extraction. Example images and descriptions can see Appendix C.

### 3.3. Text-Augmented Learner (TAL)

The TAL module provides contextual textual representations, either pre-defined (e.g., expert annotations) or dynamically generated, offering flexibility across diverse scenarios.

For dynamically generated prompts, TAL extracts key statistical properties from the input time series, including:

- **Statistical Properties:** Value range (min/max), central tendency (median), and overall trend direction.

- **Contextual Information:** Periodic description, task-specific parameters (input window length and forecasting horizon), and domain-specific dataset characteristics.

These features are formatted into structured textual prompts, such as the example shown in Figure 2(c). When domain-specific knowledge is available (e.g., in medical diagnostics or financial analysis), TAL incorporates pre-defined textual descriptions, which are combined with the dynamically generated prompts to enhance contextual understanding.

The final textual inputs are processed by the VLM text encoder, producing contextual embeddings that complement both visual and temporal features. This dual-path design supporting both static and dynamic textual ensures strong generalization across a wide range of applications, from generic forecasting tasks to specialized domain scenarios.

## 3.4. Multimodal Fusion with VLMs

The multimodal fusion pipeline integrates visual (VAL), textual (TAL), and temporal (RAL) information, leveraging their complementary strengths to enhance time series forecasting. It consists of three key steps:

**Multimodal Embeddings Extraction:** The generated images and text are processed by a frozen VLM (e.g., ViLT or CLIP), producing multimodal embeddings of shape $\mathbb{R}^{B \times L_f \times d_h}$, where $B$ is the batch size, $L_f$ is the sequence length, and $d_h$ is the VLM's hidden dimension. These embeddings capture visual and textual context, leveraging the VLM's pre-trained multimodal understanding capabilities.

**Temporal Feature Fusion:** To address the distribution shift between temporal and multimodal features, both modalities are projected into a shared $d_{\text{model}}$-dimensional space. Temporal memory embeddings $\mathbf{F}_{\text{tem}}$ from RAL encode high-level temporal patterns and serve as queries in a cross-modal multi-head attention (CM-MHA) mechanism, while the multimodal embeddings $\mathbf{F}_{\text{mm}}$ from the VLM serve as keys and values. The CM-MHA is defined as:

$$\text{CM-MHA}(Q, K, V) = \text{Cat}(\text{head}_1, \ldots, \text{head}_h)W^O, \quad (10)$$

$$\text{head}_i = \text{softmax}\left(\frac{QW_i^Q(KW_i^K)^\top}{\sqrt{d_k}}\right)VW_i^V. \quad (11)$$

where $Q = \mathbf{F}_{\text{tem}}W^Q$, $K = \mathbf{F}_{\text{mm}}W^K$, and $V = \mathbf{F}_{\text{mm}}W^V$. Here, $W_i^Q$, $W_i^K$, $W_i^V$, and $W^O$ are learnable projection matrices. $d_k = d_{\text{model}}/h$ is the head dimension, and $h$ is the number of attention heads. This mechanism aligns and integrates temporal and multimodal features, capturing both fine-grained patterns and high-level context. A residual connection and layer normalization stabilize training:

$$\mathbf{F}_{\text{attn}} = \text{LayerNorm}(\mathbf{F}_{\text{tem}} + \text{CM-MHA}(Q, K, V)). \quad (12)$$

A gated fusion mechanism further enhances the output by dynamically weighting each modality:

$$\mathbf{G} = \sigma(\mathbf{W}_g[\mathbf{F}_{\text{tem}}; \mathbf{F}_{\text{mm}}] + \mathbf{b}_g), \quad (13)$$

$$\mathbf{F}_{\text{fused}} = \mathbf{G} \odot \mathbf{F}_{\text{attn}} + (1 - \mathbf{G}) \odot \mathbf{F}_{\text{mm}}, \quad (14)$$

where $\mathbf{W}_g$ and $\mathbf{b}_g$ are learnable parameters, and $\sigma(\cdot)$ is the sigmoid function. This gated mechanism adaptively balances temporal and multimodal features for robust fusion.

**Forecasting:** The fused embedding is processed by a fine-tuned predictor, consisting of fully connected layers, to generate forecasts $\hat{y} \in \mathbb{R}^{B \times T_{\text{pred}} \times D}$. By combining visual, textual, and temporal modalities, the pipeline captures both detailed patterns and high-level context, leveraging pre-trained VLMs for enhanced forecasting across diverse scenes.

## 3.5. Optimization

The model is trained end-to-end using mean squared error (MSE). Given historical observations $\mathbf{X} \in \mathbb{R}^{N \times T}$ (with $N$ variables and $T$ time steps), the objective is to predict future values $\hat{\mathbf{Y}} \in \mathbb{R}^{N \times H}$ over $H$ steps:

$$\mathcal{L} = \frac{1}{H}\sum_{h=1}^{H}\|\hat{\mathbf{Y}}_h - \mathbf{Y}_h\|^2, \quad (15)$$

where $\hat{\mathbf{Y}}_h$ and $\mathbf{Y}_h$ denote predicted and ground-truth values at step $h$. The pre-trained VLM is kept frozen, and only lightweight components are optimized during fine-tuning:

- **RAL:** Patch embedding, memory retrieval, and attention modules for temporal pattern learning;
- **VAL:** Frequency/periodicity encoding and multi-scale CNNs for visual representation generation;
- **Prediction Head:** A gate network and linear projection fuse multimodal features to generate final forecasts.

This strategy adapts the VLM to time series forecasting, achieving robust performance with minimal overhead.

## 4. Experiments

**Datasets and Metrics.** We evaluate Time-VLM on seven widely used time series datasets across diverse domains: energy (ETTh1, ETTh2, ETTm1, ETTm2), weather, electricity (ECL), and traffic (Zhou et al., 2021; Lai et al., 2018). These datasets are commonly used for benchmarking long-term forecasting models (Wu et al., 2023a), and vary in frequency, dimensionality, and temporal characteristics. For short-term forecasting, we use the M4 benchmark (Makridakis et al., 2018), which includes marketing data across multiple frequencies. Performance is measured using Mean Absolute Error (MAE) and Mean Squared Error (MSE), following standard evaluation practices in this field. Additional details are provided in Appendices A.1 and A.3.

**Baselines.** We compare Time-VLM with state-of-the-art time series models, including text-augmented methods like TimeLLM (2024), GPT4TS (2023), and LLMTime (2023); vision-augmented methods like TimesNet (2023b); traditional deep models like PatchTST (2023), ESTformer (2022), Non-Stationary Transformer (2022b), FEDformer (2022), Autoformer (2021), Informer (2021), and Reformer (2020); and recent competitive models like DLinear (2023), LightTS (2022), N-HiTS (2023), and N-BEATS (2020). Notably, Time-VLM is the first framework combining three modalities for time series forecasting. Performance results for some baselines are cited from (2024a) where applicable.

**Implementation Details.** We compare Time-VLM against superior baselines using a unified evaluation pipeline under the same configurations as (Wu et al., 2023a) to ensure a fair comparison. ViLT (Kim et al., 2021)

Table 1: Few-shot learning on 5% training data. Results are averaged over forecasting horizons $H \in \{96, 192, 336, 720\}$. Lower values indicate better performance. Full results see Section B.1. **Red**: best, Blue: second best.

| Methods | Time-VLM_143M (Ours) | | Time-LLM_3405M (2024) | | GPT4TS (2023) | | DLinear (2023) | | PatchTST (2023) | | TimesNet (2023a) | | FEDformer (2022) | | Autoformer (2021) | | Stationary (2022b) | | ETSformer (2022) | | LightTS (2022) | | Informer (2021) | | Reformer (2020) | |
|---|---|---|---|---|---|---|---|---|---|---|---|---|---|---|---|---|---|---|---|---|---|---|---|---|---|---|
| Metric | MSE | MAE | MSE | MAE | MSE | MAE | MSE | MAE | MSE | MAE | MSE | MAE | MSE | MAE | MSE | MAE | MSE | MAE | MSE | MAE | MSE | MAE | MSE | MAE | MSE | MAE |
| $ETTh1$ | **0.442** | **0.453** | 0.627 | 0.543 | 0.681 | 0.560 | 0.750 | 0.611 | 0.694 | 0.569 | 0.925 | 0.647 | 0.658 | 0.562 | 0.722 | 0.598 | 0.943 | 0.646 | 1.189 | 0.839 | 1.451 | 0.903 | 1.225 | 0.817 | 1.241 | 0.835 |
| $ETTh2$ | **0.354** | **0.402** | 0.382 | 0.418 | 0.400 | 0.433 | 0.694 | 0.577 | 0.827 | 0.615 | 0.439 | 0.448 | 0.463 | 0.454 | 0.441 | 0.457 | 0.470 | 0.489 | 0.809 | 0.681 | 3.206 | 1.268 | 3.922 | 1.653 | 3.527 | 1.472 |
| $ETTm1$ | **0.364** | **0.385** | 0.425 | 0.434 | 0.472 | 0.450 | 0.400 | 0.417 | 0.526 | 0.476 | 0.717 | 0.561 | 0.730 | 0.592 | 0.796 | 0.620 | 0.857 | 0.598 | 1.125 | 0.782 | 1.123 | 0.765 | 1.163 | 0.791 | 1.264 | 0.826 |
| $ETTm2$ | **0.262** | **0.323** | 0.274 | 0.323 | 0.308 | 0.346 | 0.399 | 0.426 | 0.314 | 0.352 | 0.344 | 0.372 | 0.381 | 0.404 | 0.388 | 0.433 | 0.341 | 0.372 | 0.534 | 0.547 | 1.415 | 0.871 | 3.658 | 1.489 | 3.581 | 1.487 |
| $Weather$ | **0.240** | **0.280** | 0.260 | 0.309 | 0.263 | 0.301 | 0.263 | 0.308 | 0.269 | 0.303 | 0.298 | 0.318 | 0.309 | 0.353 | 0.310 | 0.353 | 0.327 | 0.328 | 0.333 | 0.371 | 0.305 | 0.345 | 0.584 | 0.527 | 0.447 | 0.453 |
| $ECL$ | 0.218 | 0.315 | 0.179 | **0.268** | **0.178** | 0.273 | 0.176 | 0.275 | 0.181 | 0.277 | 0.402 | 0.453 | 0.266 | 0.353 | 0.346 | 0.404 | 0.627 | 0.603 | 0.800 | 0.685 | 0.878 | 0.725 | 1.281 | 0.929 | 1.289 | 0.904 |
| $Traffic$ | 0.558 | 0.410 | 0.423 | 0.298 | 0.434 | 0.305 | 0.450 | 0.317 | **0.418** | **0.296** | 0.867 | 0.493 | 0.676 | 0.423 | 0.833 | 0.502 | 1.526 | 0.839 | 1.859 | 0.927 | 1.557 | 0.795 | 1.591 | 0.832 | 1.618 | 0.851 |

Table 2: Few-shot learning on 10% training data. We use the same protocol in Table 1. Full results see Section B.1.

| Methods | Time-VLM_143M (Ours) | | Time-LLM_3405M (2024) | | GPT4TS (2023) | | DLinear (2023) | | PatchTST (2023) | | TimesNet (2023a) | | FEDformer (2022) | | Autoformer (2021) | | Stationary (2022b) | | ETSformer (2022) | | LightTS (2022) | | Informer (2021) | | Reformer (2020) | |
|---|---|---|---|---|---|---|---|---|---|---|---|---|---|---|---|---|---|---|---|---|---|---|---|---|---|---|
| Metric | MSE | MAE | MSE | MAE | MSE | MAE | MSE | MAE | MSE | MAE | MSE | MAE | MSE | MAE | MSE | MAE | MSE | MAE | MSE | MAE | MSE | MAE | MSE | MAE | MSE | MAE |
| $ETTh1$ | **0.431** | **0.442** | 0.556 | 0.522 | 0.590 | 0.525 | 0.691 | 0.600 | 0.633 | 0.542 | 0.869 | 0.628 | 0.639 | 0.561 | 0.702 | 0.596 | 0.915 | 0.639 | 1.180 | 0.834 | 1.375 | 0.877 | 1.199 | 0.809 | 1.249 | 0.833 |
| $ETTh2$ | **0.361** | 0.405 | 0.370 | **0.394** | 0.397 | 0.421 | 0.605 | 0.538 | 0.415 | 0.431 | 0.479 | 0.465 | 0.466 | 0.475 | 0.488 | 0.499 | 0.462 | 0.455 | 0.894 | 0.713 | 2.655 | 1.160 | 3.872 | 1.513 | 3.485 | 1.486 |
| $ETTm1$ | **0.360** | **0.382** | 0.404 | 0.427 | 0.464 | 0.441 | 0.411 | 0.429 | 0.501 | 0.466 | 0.677 | 0.537 | 0.722 | 0.605 | 0.802 | 0.628 | 0.797 | 0.578 | 0.980 | 0.714 | 0.971 | 0.705 | 1.192 | 0.821 | 1.426 | 0.856 |
| $ETTm2$ | **0.263** | **0.323** | 0.277 | 0.323 | 0.293 | 0.335 | 0.316 | 0.368 | 0.296 | 0.343 | 0.320 | 0.353 | 0.463 | 0.488 | 1.342 | 0.930 | 0.332 | 0.366 | 0.447 | 0.487 | 0.987 | 0.756 | 3.370 | 1.440 | 3.978 | 1.587 |
| $Weather$ | **0.233** | 0.274 | 0.234 | **0.273** | 0.238 | 0.275 | 0.241 | 0.283 | 0.242 | 0.279 | 0.279 | 0.301 | 0.284 | 0.324 | 0.300 | 0.342 | 0.318 | 0.323 | 0.318 | 0.360 | 0.289 | 0.322 | 0.597 | 0.495 | 0.546 | 0.469 |
| $ECL$ | 0.198 | 0.291 | **0.175** | 0.270 | 0.176 | 0.269 | 0.180 | 0.280 | 0.180 | 0.273 | 0.323 | 0.392 | 0.346 | 0.427 | 0.431 | 0.478 | 0.444 | 0.480 | 0.660 | 0.617 | 0.441 | 0.489 | 1.195 | 0.891 | 0.965 | 0.768 |
| $Traffic$ | 0.484 | 0.357 | 0.429 | 0.306 | 0.440 | 0.310 | 0.447 | 0.313 | 0.430 | **0.305** | 0.951 | 0.535 | 0.663 | 0.425 | 0.749 | 0.446 | 1.453 | 0.815 | 1.914 | 0.936 | 1.248 | 0.684 | 1.534 | 0.811 | 1.551 | 0.821 |

Table 3: Zero-shot learning results. Full results see Section B.2.

| Methods | Time-VLM_143M (Ours) | | Time-LLM_3405M (2024) | | LLMTime (2023) | | GPT4TS (2023) | | DLinear (2023) | | PatchTST (2023) | |
|---|---|---|---|---|---|---|---|---|---|---|---|---|
| Metric | MSE | MAE | MSE | MAE | MSE | MAE | MSE | MAE | MSE | MAE | MSE | MAE |
| $ETTh1 \rightarrow ETTh2$ | **0.338** | **0.385** | 0.353 | 0.387 | 0.992 | 0.708 | 0.406 | 0.422 | 0.493 | 0.488 | 0.380 | 0.405 |
| $ETTh1 \rightarrow ETTm2$ | 0.293 | 0.350 | **0.273** | **0.340** | 1.867 | 0.869 | 0.325 | 0.363 | 0.415 | 0.452 | 0.314 | 0.360 |
| $ETTh2 \rightarrow ETTh1$ | 0.496 | 0.480 | **0.479** | **0.474** | 1.961 | 0.981 | 0.757 | 0.578 | 0.703 | 0.574 | 0.565 | 0.513 |
| $ETTh2 \rightarrow ETTm2$ | 0.297 | 0.353 | **0.272** | **0.341** | 1.867 | 0.869 | 0.335 | 0.370 | 0.328 | 0.386 | 0.325 | 0.365 |
| $ETTm1 \rightarrow ETTh2$ | **0.354** | **0.397** | 0.381 | 0.412 | 0.992 | 0.708 | 0.433 | 0.439 | 0.464 | 0.475 | 0.439 | 0.438 |
| $ETTm1 \rightarrow ETTm2$ | **0.264** | **0.319** | 0.268 | 0.320 | 1.867 | 0.869 | 0.313 | 0.348 | 0.335 | 0.389 | 0.296 | 0.334 |
| $ETTm2 \rightarrow ETTh2$ | 0.359 | 0.399 | **0.354** | **0.400** | 0.992 | 0.708 | 0.435 | 0.443 | 0.455 | 0.471 | 0.409 | 0.425 |
| $ETTm2 \rightarrow ETTm1$ | 0.432 | 0.426 | **0.414** | 0.438 | 1.933 | 0.984 | 0.769 | 0.567 | 0.649 | 0.537 | 0.568 | 0.492 |

("vilt-b32-finetuned-coco") serves as the default vision-language backbone; CLIP and BLIP-2 are also supported. All models are trained with Adam ($10^{-3}$ initial learning rate, halved per epoch), batch size 32, for up to 10 epochs with early stopping. Experiments run on Nvidia RTX A6000 GPU (48GB). More details are in Appendix A.2.

## 4.1. Few-shot Forecasting

**Setting.** We evaluate the few-shot long-term forecasting capabilities of Time-VLM by testing its performance using only 5% or 10% of the training data. This setting assesses how effectively Time-VLM integrates pre-trained multimodal knowledge from the VLM with time series-specific features under minimal task-specific supervision.

**Results.** As shown in Table 1 and Table 2, Time-VLM consistently outperforms most baselines across datasets. For example, on ETTh1 with 5% training data, Time-VLM reduces MSE by 29.5% and MAE by 16.6% compared to the second-best model, TimeLLM. On ETTm1 with 10% data, it surpasses TimeLLM by 11.1% in MSE and 10.5% in

MAE. On Weather with 5% data, Time-VLM outperforms TimeLLM by 7.7% in MSE and 9.4% in MAE. The performance gap between Time-VLM and traditional models (e.g., PatchTST, FEDformer) is particularly pronounced in few-shot settings, demonstrating the effectiveness of multimodal integration when data is scarce. This performance gain stems from the model's ability to leverage rich multimodal priors from pre-trained VLMs, while effectively capturing temporal patterns through memory-enhanced attention.

## 4.2. Zero-shot Forecasting

**Setting.** We evaluate the zero-shot forecasting capability of Time-VLM in cross-domain settings, where the model predicts on unseen datasets by effectively transferring knowledge from unrelated domains. To ensure a rigorous comparison, we conduct experiments using the ETT datasets as source and target domains, following previous setup (Jin et al., 2024). Results are summarized in Table 3.

**Results.** Time-VLM demonstrates strong generalizability, consistently outperforming or matching state-of-the-art baselines while using fewer parameters. For example, in the ETTh1->ETTh2 transfer setting, Time-VLM achieves a 4.2% lower MSE and 0.5% lower MAE than TimeLLM. In ETTm1->ETTh2, it outperforms TimeLLM by 7.1% in MSE and 3.6% in MAE. In ETTm2->ETTh2, Time-VLM performs competitively, closely matching TimeLLM with only a 1.4% difference in MSE and 0.3% in MAE. These results highlight Time-VLM's ability to generalize across domains without fine-tuning, leveraging pre-trained vision-language priors for effective knowledge transfer.

Table 4: Short-term time series forecasting results on M4. The forecasting horizons are in [6, 48] and the three rows provided are weighted averaged from all datasets under different sampling intervals. Full results see Section B.3.

| Methods | Time-VLM_143M (Ours) | Time-LLM_3405M (2024) | GPT4TS (2023) | TimesNet (2023a) | PatchTST (2023) | N-HiTS (2023) | N-BEATS (2020) | ETSformer (2022) | LightTS (2022) | DLinear (2023) | FEDformer (2022) | Stationary (2022b) | Autoformer (2021) | Informer (2021) | Reformer (2020) |
|---|---|---|---|---|---|---|---|---|---|---|---|---|---|---|---|
| SMAPE | **11.894** | 11.983 | 12.690 | 12.880 | 12.059 | 12.035 | 12.250 | 14.718 | 13.525 | 13.639 | 13.160 | 12.780 | 12.909 | 14.086 | 18.200 |
| MASE | **1.592** | 1.595 | 1.808 | 1.836 | 1.623 | 1.625 | 1.698 | 2.408 | 2.111 | 2.095 | 1.775 | 1.756 | 1.771 | 2.718 | 4.223 |
| OWA | **0.855** | 0.859 | 0.940 | 0.955 | 0.869 | 0.869 | 0.896 | 1.172 | 1.051 | 1.051 | 0.949 | 0.930 | 0.939 | 1.230 | 1.775 |

Table 5: Long-term forecasting results. We use the same protocol in Table 1. Full results see in Section B.4.

| Methods | Time-VLM_143M (Ours) | | Time-LLM_3405M (2024) | | GPT4TS (2023) | | DLinear (2023) | | PatchTST (2023) | | TimesNet (2023a) | | FEDformer (2022) | | Autoformer (2021) | | Stationary (2022b) | | ETSformer (2022) | | LightTS (2022) | | Informer (2021) | | Reformer (2020) | |
|---|---|---|---|---|---|---|---|---|---|---|---|---|---|---|---|---|---|---|---|---|---|---|---|---|---|---|---|
| Metric | MSE | MAE | MSE | MAE | MSE | MAE | MSE | MAE | MSE | MAE | MSE | MAE | MSE | MAE | MSE | MAE | MSE | MAE | MSE | MAE | MSE | MAE | MSE | MAE | MSE | MAE |
| ETTh1 | **0.405** | **0.420** | 0.408 | 0.423 | 0.465 | 0.455 | 0.422 | 0.437 | 0.413 | 0.430 | 0.458 | 0.450 | 0.440 | 0.460 | 0.496 | 0.487 | 0.570 | 0.537 | 0.542 | 0.510 | 0.491 | 0.479 | 1.040 | 0.795 | 1.029 | 0.805 |
| ETTh2 | 0.341 | 0.391 | 0.334 | 0.383 | 0.381 | 0.412 | 0.431 | 0.446 | **0.330** | **0.379** | 0.414 | 0.427 | 0.437 | 0.449 | 0.450 | 0.459 | 0.526 | 0.516 | 0.439 | 0.452 | 0.602 | 0.543 | 4.431 | 1.729 | 6.736 | 2.191 |
| ETTm1 | 0.347 | **0.377** | 0.329 | 0.372 | 0.388 | 0.403 | 0.357 | 0.378 | 0.351 | 0.380 | 0.400 | 0.406 | 0.448 | 0.452 | 0.588 | 0.517 | 0.481 | 0.456 | 0.429 | 0.425 | 0.435 | 0.437 | 0.961 | 0.734 | 0.799 | 0.671 |
| ETTm2 | **0.248** | **0.311** | 0.251 | 0.313 | 0.284 | 0.339 | 0.267 | 0.333 | 0.255 | 0.315 | 0.291 | 0.333 | 0.305 | 0.349 | 0.327 | 0.371 | 0.306 | 0.347 | 0.293 | 0.342 | 0.409 | 0.436 | 1.410 | 0.810 | 1.479 | 0.915 |
| Weather | **0.224** | 0.263 | 0.225 | 0.257 | 0.237 | 0.270 | 0.248 | 0.300 | 0.225 | 0.264 | 0.259 | 0.287 | 0.309 | 0.360 | 0.338 | 0.382 | 0.288 | 0.314 | 0.271 | 0.334 | 0.261 | 0.312 | 0.634 | 0.548 | 0.803 | 0.656 |
| Electricity | 0.172 | 0.273 | **0.158** | **0.252** | 0.167 | 0.263 | 0.166 | 0.263 | 0.161 | 0.252 | 0.192 | 0.295 | 0.214 | 0.327 | 0.227 | 0.338 | 0.193 | 0.296 | 0.208 | 0.323 | 0.229 | 0.329 | 0.311 | 0.397 | 0.338 | 0.422 |
| Traffic | 0.419 | 0.303 | **0.388** | 0.264 | 0.414 | 0.294 | 0.433 | 0.295 | **0.390** | **0.263** | 0.620 | 0.336 | 0.610 | 0.376 | 0.628 | 0.379 | 0.624 | 0.340 | 0.621 | 0.396 | 0.622 | 0.392 | 0.764 | 0.416 | 0.741 | 0.422 |

## 4.3. Short-term Forecasting

**Setting.** For short-term forecasting, we evaluate Time-VLM on the M4 benchmark, which includes marketing data at various sampling frequencies. Performance is measured using SMAPE, MASE, and OWA metrics, averaged across datasets and sampling intervals (see Table 4).

**Results.** Time-VLM demonstrates strong performance, consistently outperforming state-of-the-art baselines across all metrics. For instance, it surpasses the second-best model, Time-LLM, with improvements of 0.7% in SMAPE, 0.2% in MASE, and 0.5% in OWA, all while utilizing significantly fewer parameters and computational resources. Compared to traditional models like PatchTST and N-HiTS, the performance gains more, highlighting the benefit of multimodal knowledge in short-term forecasting. These gains stem from Time-VLM's integration of temporal, visual, and textual data, capturing richer features for improved accuracy.

## 4.4. Long-term Forecasting

**Setting.** We evaluate the long-term forecasting capabilities of Time-VLM across multiple horizons and datasets.

**Results.** As shown in Table 5, Time-VLM achieves competitive performance compared to state-of-the-art baselines. On ETTh1, Time-VLM improves upon TimeLLM by 0.7% in both MSE and MAE. On ETTm2, it outperforms TimeLLM by 1.2% in MSE and 0.6% in MAE. However, on the Weather dataset, Time-VLM slightly underperforms TimeLLM, with a 0.4% higher MSE and 2.3% higher MAE.

Overall, Time-VLM demonstrates robust performance across diverse tasks and datasets, highlighting its generalization and efficiency. By leveraging multimodal knowledge, it consistently outperforms state-of-the-art baselines with significantly fewer parameters (143M vs. TimeLLM's 3405M), making it a practical solution for real-world applications.

## 4.5. Model Analysis

**Ablation Studies.** Table 6 evaluates the contributions of key components of Time-VLM, including the RAL (with Local and Global Memory), VAL, and TAL. The study highlights the importance of each module. Removing the RAL causes a significant performance drop (35.6% in MSE), with its local (RAL_L) and global (RAL_G) branches contributing 17.2% and 4.3%, respectively, validating our hierarchical memory design. The VAL is also essential—removing it increases MSE by 9.0%, demonstrating its ability to preserve fine-grained temporal patterns via the VLM vision encoder. In contrast, removing the TAL results in only minor degradation (2.1% in MSE), likely due to the sparsity of textual tokens in the VLM output; for example, ViLT produces just 11 textual tokens out of 156 total, the rest being visual embeddings. While the TAL provides useful semantic context, its impact is limited by the scarcity of textual signals. Future work may explore VLMs with stronger language capabilities for better temporal-semantic alignment.

Table 6: Ablation study on multimodal components over forecasting horizons $H \in \{96, 192, 336, 720\}$ on Weather dataset, with MSE performance degradation (*%Deg*) measured for each variant.

| Horizon | Full | w/o RAL | w/o RAL_L | w/o RAL_G | w/o VAL | w/o TAL |
|---|---|---|---|---|---|---|
| 96 | **0.160** | 0.273 | 0.185 | 0.165 | 0.213 | 0.165 |
| 192 | **0.203** | 0.297 | 0.235 | 0.210 | 0.237 | 0.208 |
| 336 | **0.253** | 0.325 | 0.295 | 0.265 | 0.255 | 0.258 |
| 720 | **0.317** | 0.369 | 0.375 | 0.330 | 0.309 | 0.322 |
| Avg | **0.233** | 0.316 | 0.273 | 0.243 | 0.254 | 0.238 |
| %Deg | – | 35.6% ↑ | 17.2% ↑ | 4.3% ↑ | 9.0% ↑ | 2.1% ↑ |

**Multimodal and Few-/Zero-shot Analysis.** To understand the source of Time-VLM's strong performance in data-scarce scenarios, we examine the relationship between RAL (temporal) and TAL/VAL (multimodal) embeddings. As shown in Figure 3, their complementary nature is evident. The left panel illustrates balanced gate weight distributions, indicating effective fusion of temporal and multimodal sig-

nals. The right panel presents a UMAP visualization revealing distinct yet partially overlapping clusters, confirming successful integration of multimodal information while preserving modality-specific characteristics. The robust few-shot and zero-shot capabilities of Time-VLM stem from its integration of temporal, visual, and textual modalities. Specifically, the RAL models temporal dependencies via memory bank interactions, enabling robust feature extraction even with limited data. The VAL captures interpretable visual patterns—such as trends, seasonality, and periodicity—in domain-agnostic representations, while the TAL generates semantic descriptions that enhance generalization. Together, these components allow Time-VLM to leverage pre-trained multimodal knowledge, making it highly adaptable to new tasks and domains with minimal fine-tuning.

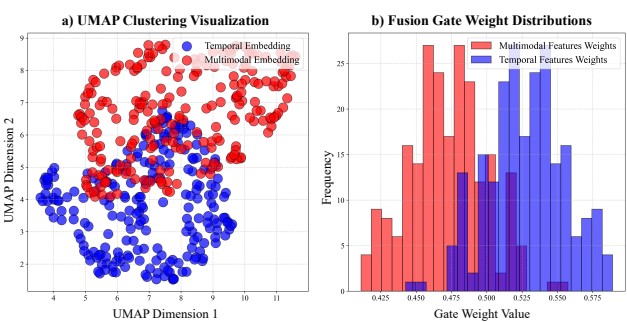

Figure 3: UMAP visualization (left) and gate weight distributions (right) of of multimodal and temporal embeddings.

**Interpretability Analysis.** We investigate how pre-trained VLMs can be leveraged for time series forecasting by analyzing the alignment between visual, textual, and temporal representations. To this end, we sample 400 image-text pairs from MSCOCO, the primary pre-training dataset for VLMs, and 200 samples each from time series datasets: ETT, Traffic, Weather, and ECL. Using UMAP, we visualize four types of embeddings in a shared 2D space:

- multimodal embeddings derived from COCO-Pair samples using VLM, reflecting the model's general pretrained cross-modal knowledge, which is Time-VLM's main motivation of the time series project to.

- multimodal embeddings from time series-generated image-text pairs processed through the same VLM, representing Time-VLM's task-specific augmentation.

- Visual-only embeddings from COCO-Image samples extracted via ViT, reflecting pure visual knowledge.

- Text-only embeddings from COCO-Text samples encoded with BERT, capturing linguistic knowledge.

As shown in Figure 4, both COCO-Image and COCO-Text form distinct, separate clusters, remaining isolated from time-series features. This suggests that while low-level visual patterns in COCO-Image resemble temporal dynamics,

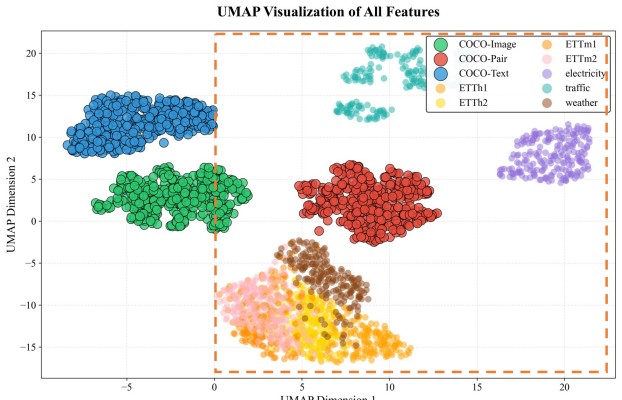

Figure 4: Interpretability visualization of Time-VLM: multimodal feature alignment via UMAP.

pure image representations retain modality-specific characteristics. Similarly, COCO-Text forms a completely separate cluster, highlighting significant modality gaps. In contrast, COCO-Pair exhibits maximal overlap with time-series data, demonstrating strong cross-modal complementarity. The textual semantics in COCO-Pair bridge visual and temporal modalities, enhancing their alignment. Notably, COCO-Pair is positioned near the center of time-series clusters, suggesting its role as a key mediator between modalities.

These observations motivate Time-VLM's design: instead of relying on single-modality projections, we embed time series into multimodal space for richer semantic understanding. The model's strong performance in data-scarce settings stems from pre-trained VLM knowledge. With more time series data, better alignment of multimodal embeddings is expected, further enhancing performance.

**Computation Studies.** Time-VLM demonstrates strong computational efficiency, as shown in Table 7. With only 143.6M parameters (1/20 of Time-LLM's 3404.6M), memory usage scales from 1968 MiB (Weather) to 24916 MiB (Traffic), adapting to dataset complexity. Inference speed ranges from 0.2057s/iter (ECL) to 0.4809s/iter (ETTh1), efficiently handling varying loads. In contrast, Time-LLM requires over 37GB of memory even for smaller datasets like ETTh1 and ETTh2, making it infeasible for larger datasets such as Weather, ECL, and Traffic. This highlights Time-VLM's lightweight design and practical scalability.

Table 7: Computational efficiency comparison between Time-VLM and Time-LLM across datasets. "-" denotes memory exceeds 49GB, infeasible on a single GPU. Results are averaged over multiple prediction steps under consistent conditions.

| Method | Metric | ETTh1 | ETTh2 | ETTm1 | ETTm2 | Weather | ECL | Traffic |
|---|---|---|---|---|---|---|---|---|
| Time-VLM | Param. (M) | 143.6 | 143.6 | 143.6 | 143.6 | 143.6 | 143.6 | 143.6 |
| | Mem. (MiB) | 2630 | 2630 | 2640 | 2640 | 1968 | 10818 | 24916 |
| | Speed (s/iter) | 0.481 | 0.438 | 0.277 | 0.210 | 0.296 | 0.206 | 0.323 |
| Time-LLM | Param. (M) | 3404.6 | 3404.6 | 3404.6 | 3404.6 | - | - | - |
| | Mem. (MiB) | 37723 | 37723 | 37849 | 37849 | - | - | - |
| | Speed (s/iter) | 0.607 | 0.553 | 0.349 | 0.265 | - | - | - |

**Hyperparameter Studies.** We analyze the impact of key hyperparameters on performance, as shown in Figure 5. Sequence length performs best between 96 and 1024 timesteps, with 512 being optimal for most datasets. Longer input introduce noise without significant gains, indicating that local temporal patterns are sufficient for accurate forecasting. The normalization constant peaks at 0.4, reflecting a balance between feature scaling and training stability. Model dimension shows dataset-dependent behavior: values of 128–256 suffice for short-term datasets like ETTh1 and ETTh2, while longer horizons and higher-dimensional data (e.g., Traffic, Weather) benefit from larger dimensions (up to 512), suggesting greater capacity is needed to model complex dynamics and variable interactions. Similarly, the gate network dimension—responsible for multimodal fusion—achieves optimal performance at 256 for medium-range forecasts. For more challenging settings like long-horizon or high-variable inputs, increasing it to 336 or 512 further improves results, highlighting the importance of adaptive fusion in capturing complex cross-modal relationships.

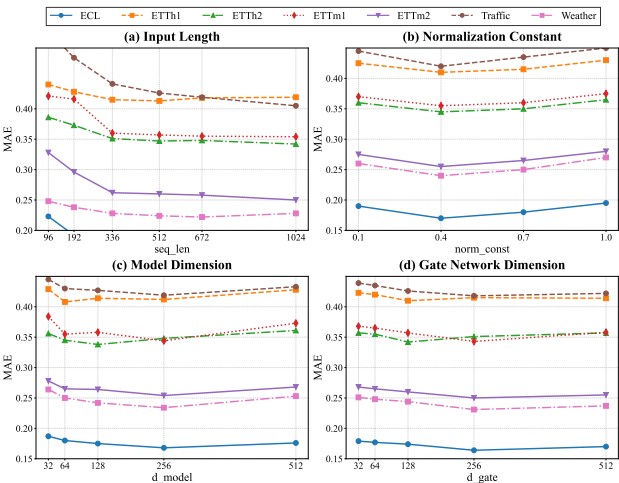

Figure 5: Hyperparameters sensitivity analysis on input length, normalization constant, dimension of model and dimension of gate network, reflected by MAE.

**VLM Variants Analysis.** We conduct ablation studies on different VLM backbones and custom combinations to assess their impact on forecasting performance and computational efficiency. We evaluate three widely-used VLMs—ViLT, CLIP, and BLIP-2—with varying size, and find that increased model size does not improve accuracy. Notably, although BLIP-2 is the largest (3.7B parameters, 25GB+ memory), it underperforms ViLT and CLIP in terms of MSE (0.342 vs. 0.337) and exhibits slow training speed (0.98 s/iter), limiting its practical use. In contrast, lightweight models like ViLT (128.9M parameters, 1346 MiB memory) and CLIP (168.4M, 1174 MiB) achieve comparable or better accuracy at a fraction of the cost. To

evaluate the benefit of VLMs' pre-trained cross-modal alignment, we construct a modular baseline using separately trained vision and language encoders: ViT-B/16 and BERT-Base. As shown in Table 8, this custom combination underperforms all pretrained VLMs across metrics, achieving an average MSE of 0.348 versus 0.336 for ViLT, with no compensating gain in speed (0.17 s/iter). These findings highlight that the cross-modal alignment in VLMs offers a key inductive bias for time series modeling. Modular approaches, lacking such a unified space, fail to match this performance—demonstrating the value of pre-aligned multimodal representations for efficient fusion and forecasting.

Table 8: Comparison of different VLM variants on ETTh2 in terms of performance and computational efficiency.

| VLM Type | Params (M) | Mem. (MiB) | Speed (s/iter) | MSE (avg) | MAE (avg) |
|---|---|---|---|---|---|
| ViLT | **128.9** | 1346 | 0.36 | **0.336** | **0.388** |
| CLIP | 168.4 | **1174** | **0.12** | 0.339 | 0.391 |
| BLIP-2 | 3763.1 | 25200 | 0.98 | 0.342 | 0.393 |
| Custom | 213.2 | 1474 | 0.17 | 0.348 | 0.397 |

## 5. Conclusion

We presented Time-VLM, a novel framework that leverages pretrained VLMs to unify temporal, visual, and textual modalities for time series forecasting. By integrating the RAL, VAL, and TAL modules, Time-VLM bridges modality gaps and enables rich cross-modal interactions. Notably, it operates solely on raw time series data without requiring external information, enabling fair comparisons and demonstrating the ability to generate textual and visual representations internally for self-augmentation. This design not only improves accuracy but also highlights the framework's robustness—particularly in domains where auxiliary data is scarce. Extensive experiments show that Time-VLM achieves superior performance across diverse datasets, especially in few-shot and zero-shot settings, outperforming existing methods while maintaining computational efficiency. Our work establishes a new direction in multimodal time series forecasting by highlighting the potential of VLMs in capturing both temporal dynamics and semantic context.

**Limitations.** Despite its strengths, Time-VLM has several limitations. First, the TAL module provides semantic context but has limited impact due to current VLMs' constrained understanding of time series semantics. Second, while excelling in low-data regimes, full-shot performance slightly lags behind specialized unimodal models on certain tasks (e.g., ECL, Traffic), suggesting room for domain-specific adaptation. Third, although computationally efficient compared to LLM-based methods, deployment on resource-constrained devices remains challenging. These limitations suggest promising directions for future work, including temporally aware VLMs, improved time series imaging, visual distillation, and enhanced text encoders with stronger temporal reasoning. For details, see Appendix D.

# Acknowledgements

This work is mainly supported by the Guangdong Basic and Applied Basic Research Foundation (No. 2025A1515011994). This work is also supported by the National Natural Science Foundation of China (No. 62402414, No. 62402420), Guangzhou Municipal Science and Technology Project (No. 2023A03J0011), the Guangzhou Industrial Information and Intelligent Key Laboratory Project (No. 2024A03J0628), a grant from State Key Laboratory of Resources and Environmental Information System, Guangdong Provincial Key Lab of Integrated Communication, Sensing and Computation for Ubiquitous Internet of Things (No. 2023B1212010007), and the CCF-DiDi GAIA Collaborative Research Funds for Young Scholars.

# Impact Statement

This paper presents work whose goal is to advance the field of Machine Learning by integrating temporal, visual, and textual modalities for time series forecasting. While our approach improves accuracy and cross-domain generalization, we acknowledge potential risks such as data privacy concerns, algorithmic bias, and increased computational costs. We encourage further research into mitigating these risks to ensure responsible deployment in high-stakes applications.

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

# A. Experimental Details

## A.1. Dataset Details

Table 9: Summary of benchmark datasets. Each dataset includes multiple time series (Dim.) with varying sequence lengths, split into training, validation, and testing sets. Data are collected at different frequencies across various domains.

| Tasks | Dataset | Dim. | Series Length | Dataset Size | Frequency | Domain | Periodicity |
|---|---|---|---|---|---|---|---|
| Long-term Forecasting | ETTm1 | 7 | {96, 192, 336, 720} | (34465, 11521, 11521) | 15 min | Temperature | 96 |
| | ETTm2 | 7 | {96, 192, 336, 720} | (34465, 11521, 11521) | 15 min | Temperature | 96 |
| | ETTh1 | 7 | {96, 192, 336, 720} | (8545, 2881, 2881) | 1 hour | Temperature | 24 |
| | ETTh2 | 7 | {96, 192, 336, 720} | (8545, 2881, 2881) | 1 hour | Temperature | 24 |
| | Electricity | 321 | {96, 192, 336, 720} | (18317, 2633, 5261) | 1 hour | Electricity | 24 |
| | Traffic | 862 | {96, 192, 336, 720} | (12185, 1757, 3509) | 1 hour | Transportation | 24 |
| | Weather | 21 | {96, 192, 336, 720} | (36792, 5271, 10540) | 10 min | Weather | 144 |
| Short-term Forecasting | M4 - Yearly | 1 | 6 | (23000, 0, 23000) | Yearly | Demographic | 1 |
| | M4 - Quarterly | 1 | 8 | (24000, 0, 24000) | Quarterly | Finance | 4 |
| | M4 - Monthly | 1 | 18 | (48000, 0, 48000) | Monthly | Industry | 3 |
| | M4 - Weekly | 1 | 13 | (359, 0, 359) | Weekly | Macro | 4 |
| | M4 - Daily | 1 | 14 | (4227, 0, 4227) | Daily | Micro | 1 |
| | M4 - Hourly | 1 | 48 | (414, 0, 414) | Hourly | Other | 24 |

The benchmark datasets used in our experiments are summarized in Table 9. These datasets span diverse domains, including temperature monitoring (*ETTm1*, *ETTm2*, *ETTh1*, *ETTh2*), electricity consumption (*Electricity*), transportation (*Traffic*), and weather forecasting (*Weather*). Each dataset contains multiple time series with varying sequence lengths, split into training, validation, and testing sets. The datasets are collected at different frequencies, ranging from 15 minutes to yearly intervals, and exhibit distinct periodic patterns. For short-term forecasting, we utilize the M4 benchmark, which includes datasets with yearly, quarterly, monthly, weekly, daily, and hourly frequencies, covering domains such as finance, industry, and demographics. This diverse collection of datasets ensures a comprehensive evaluation of our method.

### A.1.1. DATASET DESCRIPTION

The datasets used in our experiments are described below:

- **ECL**: Measurements of electric power consumption in one household with a one-minute sampling rate over 4 years. It includes various electrical quantities and sub-metering values, totaling 2,075,259 measurements from a house in Sceaux, France (December 2006 to November 2010).

- **ETT**: The Electricity Transformer Temperature (ETT) dataset, crucial for electric power deployment, contains 2 years of data from two counties in China. Subsets *ETTh1* and *ETTh2* provide 1-hour-level data, while *ETTm1* offers 15-minute-level data. Each point includes the target "oil temperature" and 6 power load features, with a 12/4/4 month train/val/test split.

- **Traffic**: Hourly data from the California Department of Transportation, describing road occupancy rates measured by sensors on San Francisco Bay area freeways.

- **Weather**: Recorded every 10 minutes throughout 2020, this dataset includes 21 meteorological indicators, such as air temperature and humidity.

- **M4**: A collection of 100,000 time series from the Makridakis Forecasting Competition, including yearly, quarterly, monthly, weekly, daily, and hourly data. The training sets have minimum observations of 13 (yearly), 16 (quarterly), 42 (monthly), 80 (weekly), 93 (daily), and 700 (hourly). Forecasts required are 6 (yearly), 8 (quarterly), 18 (monthly), 13 (weekly), 14 (daily), and 48 (hourly).

### A.1.2. PERIODICITY PARAMETER

The *Periodicity* column in Table 9 specifies the periodicity hyperparameter $P$ used in the periodicity encoding process. This parameter is derived from the inherent characteristics of each dataset and reflects the dominant temporal patterns, such as

daily, weekly, or seasonal cycles. For example, in the *ETTm1* and *ETTm2* datasets, which are sampled every 15 minutes, the periodicity $P = 96$ corresponds to a daily cycle (24 hours $\times$ 4 samples per hour). Similarly, for the *ETTh1* and *ETTh2* datasets, sampled hourly, $P = 24$ represents a daily cycle. The *Weather* dataset, sampled every 10 minutes, has $P = 144$, reflecting a daily cycle (24 hours $\times$ 6 samples per hour). For the M4 benchmark datasets, the periodicity values are set based on their sampling frequencies: $P = 1$ for yearly data, $P = 4$ for quarterly and weekly data, $P = 3$ for monthly data, and $P = 24$ for hourly data. These values are used in the periodicity encoding formula:

$$\text{encoding}(t) = \left[ \sin\left(\frac{2\pi t}{P}\right), \cos\left(\frac{2\pi t}{P}\right) \right], \tag{16}$$

where $t$ is the time step and $P$ is the periodicity hyperparameter. The resulting encodings are concatenated with the input time series, enriching the model's ability to capture temporal dependencies and periodic patterns.

### A.2. Optimization Settings

#### A.2.1. MODEL ARCHITECTURE PARAMETERS

Time-VLM consists of several key components, each with specific parameter configurations. Image representations are set to a size of $64 \times 64$, balancing computational efficiency and temporal information preservation. The model backbone utilizes a hidden dimension of $d\_model = 128$, while the encoder-decoder structure comprises $e\_layers = 2$ encoder layers and $d\_layers = 1$ decoder layer. A dropout rate of $0.1$ is applied to mitigate overfitting during training. For efficient data loading, the model employs $num\_workers = 32$ to parallelize data preprocessing tasks.

The gated fusion module is designed with a dimension of $d\_fusion = 256$, facilitating the effective integration of multimodal features. The VLM component generates multimodal embeddings with a token length of $vlm\_fused\_len = 156$ and a hidden dimension of $vlm\_hidden\_dim = 768$, ensuring seamless compatibility with the pre-trained VLM's architecture.

Table 10: Default Model Architecture Parameters

| Parameter | Default Value | Description |
|---|---|---|
| image_size | 64 | Size of generated image representation |
| d_model | 128 | Dimension of hidden embeddings |
| d_fusion | 256 | Dimension of gated fusion module |
| num_workers | 32 | Number of data loader workers |
| e_layers | 2 | Number of encoder layers |
| d_layers | 1 | Number of decoder layers |
| dropout | 0.1 | Dropout rate |
| vlm_fused_len | 156 | Token length of VLM multimodal embedding |
| vlm_hidden_dim | 768 | Hidden dimension of VLM |

#### A.2.2. TRAINING PARAMETERS

We adopt a comprehensive training strategy with both general and task-specific parameters. The model is trained with a batch size of 32 and an initial learning rate of $0.001$, using the *AdamW* optimizer. Early stopping with a patience of 3 epochs is implemented to prevent overfitting. The training process employs Mean Squared Error (MSE) as the primary loss function and runs for a maximum of 10 epochs. For time series processing, we use an input sequence length of 512 and prediction lengths of 96, 192, 336, or 720, depending on the task. The output dimension ($c\_out$) varies by dataset: 7 for ETTh1/h2/m1/m2, 21 for Weather, 321 for Electricity, and 862 for Traffic. The periodicity parameter is set to 24 for ETTh1/h2, Electricity, and Traffic; 96 for ETTm1/m2; and 144 for Weather, ensuring alignment with dataset-specific temporal patterns. A normalization coefficient of $0.4$ is applied to stabilize training dynamics. The patch embedding module uses a patch length of 16, a stride of 8, and padding of 8 to process the input sequences. The temporal memory mechanism employs 8 learnable queries and 4 attention heads to capture high-level dependencies. Additionally, the training process leverages automatic mixed precision (AMP) to accelerate training while maintaining numerical stability.

Table 11: Default Training Parameters

| Parameter | Default Value | Description |
|---|---|---|
| batch_size | 32 | Training batch size |
| learning_rate | 0.001 | Initial learning rate |
| training_epochs | 10 | Number of training epochs |
| patience | 3 | Early stopping patience |
| loss | MSE | Mean square error |
| seq_len | 512 | Input sequence length |
| c_out | 7 (ETTh1/h2/m1/m2) 21 (Weather) 321 (Electricity) 862 (Traffic) | Output dimension (dataset-specific) |
| pred_len | 96/192/336/720 | Prediction length |
| periodicity | 24 (ETTh1/h2/Electricity/Traffic) 96 (ETTm1/m2) 144 (Weather) | Dataset periodicity (dataset-specific) |
| norm_const | 0.4 | Normalization coefficient |
| patch_len | 16 | Patch length |
| padding | 8 | Padding length |
| stride | 8 | Stride length |
| num_queries | 8 | Number of learnable queries for temporal memory |
| n_heads | 4 | Number of attention heads |

## A.3. Evaluation Metrics

For evaluation, we utilize mean squared error (MSE) and mean absolute error (MAE) for long-term forecasting. For short-term forecasting on the M4 benchmark, we adopt symmetric mean absolute percentage error (SMAPE), mean absolute scaled error (MASE), and overall weighted average (OWA), following the evaluation protocol of N-BEATS (Oreshkin et al., 2020). OWA is a specific metric used in the M4 competition. The metrics are calculated as follows:

$$\text{MSE} = \frac{1}{H}\sum_{h=1}^{T}(\mathbf{Y}_h - \hat{\mathbf{Y}}_h)^2, \qquad \text{MAE} = \frac{1}{H}\sum_{h=1}^{H}|\mathbf{Y}_h - \hat{\mathbf{Y}}_h|,$$

$$\text{SMAPE} = \frac{200}{H}\sum_{h=1}^{H}\frac{|\mathbf{Y}_h - \hat{\mathbf{Y}}_h|}{|\mathbf{Y}_h| + |\hat{\mathbf{Y}}_h|}, \qquad \text{MAPE} = \frac{100}{H}\sum_{h=1}^{H}\frac{|\mathbf{Y}_h - \hat{\mathbf{Y}}_h|}{|\mathbf{Y}_h|},$$

$$\text{MASE} = \frac{1}{H}\sum_{h=1}^{H}\frac{|\mathbf{Y}_h - \hat{\mathbf{Y}}_h|}{\frac{1}{H-s}\sum_{j=s+1}^{H}|\mathbf{Y}_j - \mathbf{Y}_{j-s}|}, \qquad \text{OWA} = \frac{1}{2}\left[\frac{\text{SMAPE}}{\text{SMAPE}_{\text{Naïve2}}} + \frac{\text{MASE}}{\text{MASE}_{\text{Naïve2}}}\right],$$

where $s$ is the periodicity of the time series, $H$ is the prediction horizon, and $\mathbf{Y}_h$ and $\hat{\mathbf{Y}}_h$ are the ground truth and prediction at time step $h$, respectively.

# B. Complete results

## B.1. Few-shot Forecasting

Table 12: Full few-shot learning results on 5% training data with forecasting horizons $H \in \{96, 192, 336, 720\}$. A lower value indicates better performance. '-' means that 5% time series is not sufficient to constitute a training set. **Red**: the best, Blue: the second best

| Methods | | Time-VLM | | Time-LLM | | GPT4TS | | DLinear | | PatchTST | | TimesNet | | FEDformer | | Autoformer | | Stationary | | ETSformer | | LightTS | | Informer | | Reformer | |
|---|---|---|---|---|---|---|---|---|---|---|---|---|---|---|---|---|---|---|---|---|---|---|---|---|---|---|---|
| Metric | | MSE | MAE | MSE | MAE | MSE | MAE | MSE | MAE | MSE | MAE | MSE | MAE | MSE | MAE | MSE | MAE | MSE | MAE | MSE | MAE | MSE | MAE | MSE | MAE | MSE | MAE |
| ETTh1 | 96 | 0.417 | 0.435 | 0.483 | 0.464 | 0.543 | 0.506 | 0.547 | 0.503 | 0.557 | 0.519 | 0.892 | 0.625 | 0.593 | 0.529 | 0.681 | 0.570 | 0.952 | 0.650 | 1.169 | 0.832 | 1.483 | 0.910 | 1.225 | 0.812 | 1.198 | 0.795 |
| | 192 | 0.450 | 0.458 | 0.629 | 0.540 | 0.748 | 0.580 | 0.720 | 0.604 | 0.711 | 0.570 | 0.940 | 0.665 | 0.652 | 0.563 | 0.725 | 0.602 | 0.943 | 0.645 | 1.221 | 0.853 | 1.525 | 0.930 | 1.249 | 0.828 | 1.273 | 0.853 |
| | 336 | 0.460 | 0.465 | 0.768 | 0.626 | 0.754 | 0.595 | 0.984 | 0.727 | 0.816 | 0.619 | 0.945 | 0.653 | 0.731 | 0.594 | 0.761 | 0.624 | 0.935 | 0.644 | 1.179 | 0.832 | 1.347 | 0.870 | 1.202 | 0.811 | 1.254 | 0.857 |
| | 720 | - | - | - | - | - | - | - | - | - | - | - | - | - | - | - | - | - | - | - | - | - | - | - | - | - | - |
| | Avg | 0.442 | 0.453 | 0.627 | 0.543 | 0.681 | 0.560 | 0.750 | 0.611 | 0.694 | 0.569 | 0.925 | 0.647 | 0.658 | 0.562 | 0.722 | 0.598 | 0.943 | 0.646 | 1.189 | 0.839 | 1.451 | 0.903 | 1.225 | 0.817 | 1.241 | 0.835 |
| ETTh2 | 96 | 0.302 | 0.365 | 0.336 | 0.397 | 0.376 | 0.421 | 0.442 | 0.456 | 0.401 | 0.421 | 0.409 | 0.420 | 0.390 | 0.424 | 0.428 | 0.468 | 0.408 | 0.423 | 0.678 | 0.619 | 2.022 | 1.006 | 3.837 | 1.508 | 3.753 | 1.518 |
| | 192 | 0.361 | 0.406 | 0.406 | 0.425 | 0.418 | 0.441 | 0.617 | 0.542 | 0.452 | 0.455 | 0.483 | 0.464 | 0.457 | 0.465 | 0.496 | 0.504 | 0.497 | 0.468 | 0.845 | 0.697 | 3.534 | 1.348 | 3.975 | 1.933 | 3.516 | 1.473 |
| | 336 | 0.398 | 0.434 | 0.405 | 0.432 | 0.408 | 0.439 | 1.424 | 0.849 | 0.464 | 0.469 | 0.499 | 0.479 | 0.477 | 0.483 | 0.486 | 0.496 | 0.507 | 0.481 | 0.905 | 0.727 | 4.063 | 1.451 | 3.956 | 1.520 | 3.312 | 1.427 |
| | 720 | - | - | - | - | - | - | - | - | - | - | - | - | - | - | - | - | - | - | - | - | - | - | - | - | - | - |
| | Avg | 0.354 | 0.402 | 0.382 | 0.418 | 0.400 | 0.433 | 0.694 | 0.577 | 0.827 | 0.615 | 0.439 | 0.448 | 0.463 | 0.454 | 0.441 | 0.457 | 0.470 | 0.489 | 0.809 | 0.681 | 3.206 | 1.268 | 3.922 | 1.653 | 3.527 | 1.472 |
| ETTm1 | 96 | 0.314 | 0.357 | 0.316 | 0.377 | 0.386 | 0.405 | 0.332 | 0.374 | 0.399 | 0.414 | 0.606 | 0.518 | 0.628 | 0.544 | 0.726 | 0.578 | 0.823 | 0.587 | 1.031 | 0.747 | 1.048 | 0.733 | 1.130 | 0.775 | 1.234 | 0.798 |
| | 192 | 0.343 | 0.373 | 0.450 | 0.464 | 0.440 | 0.438 | 0.358 | 0.390 | 0.441 | 0.436 | 0.681 | 0.539 | 0.666 | 0.566 | 0.750 | 0.591 | 0.844 | 0.591 | 1.087 | 0.766 | 1.097 | 0.756 | 1.150 | 0.788 | 1.287 | 0.839 |
| | 336 | 0.373 | 0.391 | 0.450 | 0.424 | 0.485 | 0.459 | 0.402 | 0.416 | 0.499 | 0.467 | 0.786 | 0.597 | 0.807 | 0.628 | 0.851 | 0.659 | 0.870 | 0.603 | 1.138 | 0.787 | 1.147 | 0.775 | 1.198 | 0.809 | 1.288 | 0.842 |
| | 720 | 0.425 | 0.420 | 0.483 | 0.471 | 0.577 | 0.499 | 0.511 | 0.489 | 0.767 | 0.587 | 0.796 | 0.593 | 0.822 | 0.633 | 0.857 | 0.655 | 0.893 | 0.611 | 1.245 | 0.831 | 1.200 | 0.799 | 1.175 | 0.794 | 1.247 | 0.828 |
| | Avg | 0.364 | 0.385 | 0.425 | 0.434 | 0.472 | 0.450 | 0.400 | 0.417 | 0.526 | 0.476 | 0.717 | 0.561 | 0.730 | 0.592 | 0.796 | 0.620 | 0.857 | 0.598 | 1.125 | 0.782 | 1.123 | 0.765 | 1.163 | 0.791 | 1.264 | 0.826 |
| ETTm2 | 96 | 0.169 | 0.260 | 0.174 | 0.261 | 0.199 | 0.280 | 0.236 | 0.326 | 0.206 | 0.288 | 0.220 | 0.299 | 0.229 | 0.320 | 0.232 | 0.322 | 0.238 | 0.316 | 0.404 | 0.485 | 1.108 | 0.772 | 3.599 | 1.478 | 3.883 | 1.545 |
| | 192 | 0.224 | 0.298 | 0.215 | 0.287 | 0.256 | 0.316 | 0.306 | 0.373 | 0.264 | 0.324 | 0.311 | 0.361 | 0.394 | 0.361 | 0.291 | 0.357 | 0.298 | 0.349 | 0.479 | 0.521 | 1.317 | 0.850 | 3.578 | 1.475 | 3.553 | 1.484 |
| | 336 | 0.282 | 0.338 | 0.273 | 0.330 | 0.318 | 0.353 | 0.380 | 0.423 | 0.334 | 0.367 | 0.338 | 0.366 | 0.378 | 0.427 | 0.478 | 0.517 | 0.353 | 0.380 | 0.552 | 0.555 | 1.415 | 0.879 | 3.561 | 1.473 | 3.446 | 1.460 |
| | 720 | 0.375 | 0.397 | 0.433 | 0.412 | 0.460 | 0.436 | 0.674 | 0.583 | 0.454 | 0.432 | 0.509 | 0.465 | 0.523 | 0.510 | 0.553 | 0.538 | 0.475 | 0.445 | 0.701 | 0.627 | 1.822 | 0.984 | 3.896 | 1.533 | 3.445 | 1.460 |
| | Avg | 0.262 | 0.323 | 0.274 | 0.323 | 0.308 | 0.346 | 0.399 | 0.426 | 0.314 | 0.352 | 0.344 | 0.372 | 0.381 | 0.404 | 0.388 | 0.433 | 0.341 | 0.372 | 0.534 | 0.547 | 1.415 | 0.871 | 3.658 | 1.489 | 3.581 | 1.487 |
| Weather | 96 | 0.176 | 0.231 | 0.172 | 0.263 | 0.175 | 0.230 | 0.184 | 0.242 | 0.171 | 0.224 | 0.207 | 0.253 | 0.229 | 0.309 | 0.227 | 0.299 | 0.215 | 0.252 | 0.218 | 0.295 | 0.230 | 0.285 | 0.497 | 0.497 | 0.406 | 0.435 |
| | 192 | 0.216 | 0.263 | 0.224 | 0.271 | 0.227 | 0.276 | 0.228 | 0.283 | 0.230 | 0.277 | 0.272 | 0.307 | 0.265 | 0.317 | 0.278 | 0.333 | 0.290 | 0.307 | 0.294 | 0.331 | 0.274 | 0.323 | 0.620 | 0.545 | 0.446 | 0.450 |
| | 336 | 0.264 | 0.298 | 0.282 | 0.321 | 0.286 | 0.322 | 0.279 | 0.322 | 0.294 | 0.326 | 0.313 | 0.328 | 0.353 | 0.392 | 0.351 | 0.393 | 0.353 | 0.348 | 0.359 | 0.398 | 0.318 | 0.355 | 0.649 | 0.547 | 0.465 | 0.459 |
| | 720 | 0.327 | 0.342 | 0.366 | 0.381 | 0.366 | 0.379 | 0.364 | 0.388 | 0.384 | 0.387 | 0.400 | 0.385 | 0.391 | 0.394 | 0.387 | 0.389 | 0.452 | 0.407 | 0.461 | 0.461 | 0.401 | 0.418 | 0.570 | 0.522 | 0.471 | 0.468 |
| | Avg | 0.246 | 0.284 | 0.260 | 0.309 | 0.263 | 0.301 | 0.263 | 0.308 | 0.269 | 0.303 | 0.298 | 0.318 | 0.309 | 0.353 | 0.310 | 0.353 | 0.327 | 0.328 | 0.333 | 0.371 | 0.305 | 0.345 | 0.584 | 0.527 | 0.447 | 0.453 |
| Electricity | 96 | 0.185 | 0.296 | 0.147 | 0.242 | 0.143 | 0.241 | 0.150 | 0.251 | 0.145 | 0.244 | 0.315 | 0.389 | 0.235 | 0.322 | 0.297 | 0.367 | 0.484 | 0.518 | 0.697 | 0.638 | 0.639 | 0.609 | 1.265 | 0.919 | 1.414 | 0.855 |
| | 192 | 0.194 | 0.302 | 0.158 | 0.241 | 0.159 | 0.255 | 0.163 | 0.263 | 0.163 | 0.260 | 0.318 | 0.396 | 0.247 | 0.341 | 0.308 | 0.375 | 0.501 | 0.531 | 0.718 | 0.648 | 0.772 | 0.678 | 1.298 | 0.939 | 1.240 | 0.919 |
| | 336 | 0.210 | 0.315 | 0.178 | 0.277 | 0.179 | 0.274 | 0.175 | 0.278 | 0.183 | 0.281 | 0.340 | 0.415 | 0.267 | 0.356 | 0.354 | 0.411 | 0.574 | 0.578 | 0.758 | 0.667 | 0.901 | 0.745 | 1.302 | 0.942 | 1.253 | 0.921 |
| | 720 | 0.251 | 0.346 | 0.224 | 0.312 | 0.233 | 0.323 | 0.219 | 0.311 | 0.233 | 0.323 | 0.635 | 0.613 | 0.318 | 0.394 | 0.426 | 0.466 | 0.952 | 0.786 | 1.028 | 0.788 | 1.200 | 0.871 | 1.259 | 0.919 | 1.249 | 0.921 |
| | Avg | 0.218 | 0.315 | 0.179 | 0.268 | 0.178 | 0.273 | 0.176 | 0.275 | 0.181 | 0.277 | 0.402 | 0.453 | 0.266 | 0.353 | 0.346 | 0.404 | 0.627 | 0.603 | 0.800 | 0.685 | 0.878 | 0.725 | 1.281 | 0.929 | 1.289 | 0.904 |
| Traffic | 96 | 0.550 | 0.408 | 0.414 | 0.291 | 0.419 | 0.298 | 0.427 | 0.304 | 0.404 | 0.286 | 0.854 | 0.492 | 0.670 | 0.421 | 0.795 | 0.481 | 1.468 | 0.821 | 1.643 | 0.855 | 1.157 | 0.636 | 1.557 | 0.821 | 1.586 | 0.841 |
| | 192 | 0.552 | 0.408 | 0.419 | 0.291 | 0.434 | 0.305 | 0.447 | 0.315 | 0.412 | 0.294 | 0.894 | 0.517 | 0.653 | 0.405 | 0.837 | 0.503 | 1.509 | 0.838 | 1.856 | 0.928 | 1.688 | 0.848 | 1.596 | 0.834 | 1.602 | 0.844 |
| | 336 | 0.572 | 0.414 | 0.437 | 0.314 | 0.449 | 0.313 | 0.478 | 0.333 | 0.439 | 0.310 | 0.853 | 0.471 | 0.707 | 0.445 | 0.867 | 0.523 | 1.602 | 0.860 | 2.080 | 0.999 | 1.826 | 0.903 | 1.621 | 0.841 | 1.668 | 0.868 |
| | 720 | - | - | - | - | - | - | - | - | - | - | - | - | - | - | - | - | - | - | - | - | - | - | - | - | - | - |
| | Avg | 0.558 | 0.410 | 0.423 | 0.298 | 0.434 | 0.305 | 0.450 | 0.317 | 0.418 | 0.296 | 0.867 | 0.493 | 0.676 | 0.423 | 0.833 | 0.502 | 1.526 | 0.839 | 1.859 | 0.927 | 1.557 | 0.795 | 1.591 | 0.832 | 1.618 | 0.851 |

Table 13: Full few-shot learning results on 10% training data.

| Methods | | Time-VLM | | Time-LLM | | GPT4TS | | DLinear | | PatchTST | | TimesNet | | FEDformer | | Autoformer | | Stationary | | ETSformer | | LightTS | | Informer | | Reformer | |
|---|---|---|---|---|---|---|---|---|---|---|---|---|---|---|---|---|---|---|---|---|---|---|---|---|---|---|---|
| Metric | | MSE | MAE | MSE | MAE | MSE | MAE | MSE | MAE | MSE | MAE | MSE | MAE | MSE | MAE | MSE | MAE | MSE | MAE | MSE | MAE | MSE | MAE | MSE | MAE | MSE | MAE |
| ETTh1 | 96 | 0.391 | 0.404 | 0.448 | 0.460 | 0.458 | 0.456 | 0.492 | 0.495 | 0.516 | 0.485 | 0.861 | 0.628 | 0.512 | 0.499 | 0.613 | 0.552 | 0.918 | 0.639 | 1.112 | 0.806 | 1.298 | 0.838 | 1.179 | 0.792 | 1.184 | 0.790 |
| | 192 | 0.420 | 0.431 | 0.484 | 0.483 | 0.570 | 0.516 | 0.565 | 0.538 | 0.598 | 0.524 | 0.797 | 0.593 | 0.624 | 0.555 | 0.722 | 0.598 | 0.915 | 0.629 | 1.155 | 0.823 | 1.322 | 0.854 | 1.199 | 0.806 | 1.295 | 0.850 |
| | 336 | 0.439 | 0.448 | 0.589 | 0.540 | 0.608 | 0.535 | 0.721 | 0.622 | 0.657 | 0.550 | 0.941 | 0.648 | 0.691 | 0.574 | 0.750 | 0.619 | 0.939 | 0.644 | 1.179 | 0.832 | 1.347 | 0.870 | 1.202 | 0.811 | 1.294 | 0.854 |
| | 720 | 0.476 | 0.484 | 0.700 | 0.604 | 0.725 | 0.591 | 0.986 | 0.743 | 0.762 | 0.610 | 0.877 | 0.641 | 0.728 | 0.614 | 0.721 | 0.616 | 0.887 | 0.645 | 1.273 | 0.874 | 1.534 | 0.947 | 1.217 | 0.825 | 1.223 | 0.838 |
| | Avg | 0.431 | 0.442 | 0.556 | 0.522 | 0.590 | 0.525 | 0.691 | 0.600 | 0.633 | 0.542 | 0.869 | 0.628 | 0.639 | 0.561 | 0.702 | 0.596 | 0.915 | 0.639 | 1.180 | 0.834 | 1.375 | 0.877 | 1.199 | 0.809 | 1.249 | 0.833 |
| ETTh2 | 96 | 0.284 | 0.347 | 0.275 | 0.326 | 0.331 | 0.374 | 0.357 | 0.411 | 0.353 | 0.389 | 0.378 | 0.409 | 0.382 | 0.416 | 0.413 | 0.451 | 0.389 | 0.411 | 0.678 | 0.619 | 2.022 | 1.006 | 3.837 | 1.508 | 3.788 | 1.533 |
| | 192 | 0.349 | 0.398 | 0.374 | 0.373 | 0.402 | 0.411 | 0.569 | 0.519 | 0.403 | 0.414 | 0.490 | 0.467 | 0.478 | 0.474 | 0.474 | 0.477 | 0.473 | 0.455 | 0.785 | 0.666 | 2.329 | 1.104 | 3.856 | 1.513 | 3.552 | 1.483 |
| | 336 | 0.370 | 0.412 | 0.406 | 0.429 | 0.406 | 0.433 | 0.671 | 0.572 | 0.426 | 0.441 | 0.537 | 0.494 | 0.504 | 0.501 | 0.547 | 0.543 | 0.507 | 0.480 | 0.839 | 0.694 | 2.453 | 1.122 | 3.952 | 1.526 | 3.395 | 1.526 |
| | 720 | 0.441 | 0.466 | 0.427 | 0.449 | 0.449 | 0.464 | 0.824 | 0.648 | 0.477 | 0.480 | 0.510 | 0.491 | 0.499 | 0.509 | 0.516 | 0.523 | 0.477 | 0.472 | 1.273 | 0.874 | 3.816 | 1.407 | 3.842 | 1.503 | 3.205 | 1.401 |
| | Avg | 0.361 | 0.405 | 0.370 | 0.394 | 0.397 | 0.421 | 0.605 | 0.538 | 0.415 | 0.431 | 0.479 | 0.465 | 0.466 | 0.475 | 0.488 | 0.499 | 0.462 | 0.455 | 0.894 | 0.713 | 2.655 | 1.160 | 3.872 | 1.513 | 3.485 | 1.486 |
| ETTm1 | 96 | 0.310 | 0.354 | 0.346 | 0.388 | 0.390 | 0.404 | 0.352 | 0.392 | 0.410 | 0.419 | 0.583 | 0.501 | 0.578 | 0.518 | 0.774 | 0.614 | 0.761 | 0.568 | 0.911 | 0.688 | 0.921 | 0.682 | 1.162 | 0.785 | 1.442 | 0.847 |
| | 192 | 0.340 | 0.370 | 0.373 | 0.416 | 0.429 | 0.423 | 0.382 | 0.412 | 0.437 | 0.434 | 0.630 | 0.528 | 0.617 | 0.546 | 0.754 | 0.592 | 0.781 | 0.574 | 0.955 | 0.703 | 0.957 | 0.701 | 1.172 | 0.793 | 1.444 | 0.862 |
| | 336 | 0.369 | 0.387 | 0.413 | 0.426 | 0.469 | 0.439 | 0.419 | 0.434 | 0.476 | 0.454 | 0.725 | 0.568 | 0.998 | 0.775 | 0.869 | 0.677 | 0.803 | 0.587 | 0.991 | 0.719 | 0.998 | 0.716 | 1.227 | 0.908 | 1.450 | 0.866 |
| | 720 | 0.423 | 0.417 | 0.485 | 0.476 | 0.569 | 0.498 | 0.490 | 0.477 | 0.681 | 0.556 | 0.769 | 0.549 | 0.693 | 0.579 | 0.810 | 0.630 | 0.844 | 0.581 | 1.062 | 0.747 | 1.007 | 0.719 | 1.207 | 0.797 | 1.366 | 0.850 |
| | Avg | 0.360 | 0.382 | 0.404 | 0.427 | 0.464 | 0.441 | 0.411 | 0.429 | 0.501 | 0.466 | 0.677 | 0.537 | 0.722 | 0.605 | 0.802 | 0.628 | 0.797 | 0.578 | 0.980 | 0.714 | 0.971 | 0.705 | 1.192 | 0.821 | 1.426 | 0.856 |
| ETTm2 | 96 | 0.169 | 0.260 | 0.177 | 0.261 | 0.188 | 0.269 | 0.213 | 0.303 | 0.191 | 0.274 | 0.212 | 0.285 | 0.291 | 0.399 | 0.352 | 0.454 | 0.229 | 0.308 | 0.331 | 0.430 | 0.813 | 0.688 | 3.203 | 1.407 | 4.195 | 1.628 |
| | 192 | 0.222 | 0.296 | 0.241 | 0.314 | 0.251 | 0.309 | 0.278 | 0.345 | 0.252 | 0.317 | 0.270 | 0.323 | 0.307 | 0.379 | 0.694 | 0.691 | 0.291 | 0.343 | 0.400 | 0.464 | 1.008 | 0.768 | 3.112 | 1.387 | 4.042 | 1.601 |
| | 336 | 0.278 | 0.335 | 0.274 | 0.327 | 0.307 | 0.346 | 0.338 | 0.385 | 0.306 | 0.353 | 0.323 | 0.353 | 0.543 | 0.559 | 2.408 | 1.407 | 0.348 | 0.376 | 0.469 | 0.498 | 1.031 | 0.775 | 3.255 | 1.421 | 3.963 | 1.585 |
| | 720 | 0.381 | 0.401 | 0.417 | 0.390 | 0.426 | 0.417 | 0.436 | 0.440 | 0.433 | 0.427 | 0.474 | 0.449 | 0.712 | 0.614 | 1.913 | 1.166 | 0.461 | 0.438 | 0.589 | 0.557 | 1.096 | 0.791 | 3.909 | 1.543 | 3.711 | 1.532 |
| | Avg | 0.263 | 0.323 | 0.277 | 0.323 | 0.293 | 0.335 | 0.316 | 0.368 | 0.296 | 0.343 | 0.320 | 0.353 | 0.463 | 0.488 | 1.342 | 0.930 | 0.332 | 0.366 | 0.447 | 0.487 | 0.987 | 0.756 | 3.370 | 1.440 | 3.978 | 1.587 |
| Weather | 96 | 0.174 | 0.228 | 0.161 | 0.210 | 0.163 | 0.215 | 0.171 | 0.224 | 0.165 | 0.215 | 0.184 | 0.230 | 0.188 | 0.253 | 0.221 | 0.297 | 0.192 | 0.234 | 0.199 | 0.272 | 0.217 | 0.269 | 0.374 | 0.401 | 0.335 | 0.380 |
| | 192 | 0.217 | 0.262 | 0.204 | 0.248 | 0.210 | 0.254 | 0.215 | 0.263 | 0.210 | 0.257 | 0.245 | 0.283 | 0.250 | 0.304 | 0.270 | 0.322 | 0.269 | 0.295 | 0.279 | 0.332 | 0.259 | 0.304 | 0.552 | 0.478 | 0.522 | 0.462 |
| | 336 | 0.263 | 0.296 | 0.261 | 0.302 | 0.256 | 0.292 | 0.258 | 0.299 | 0.259 | 0.297 | 0.305 | 0.321 | 0.312 | 0.346 | 0.320 | 0.351 | 0.370 | 0.357 | 0.356 | 0.386 | 0.303 | 0.334 | 0.724 | 0.541 | 0.715 | 0.535 |
| | 720 | 0.326 | 0.340 | 0.309 | 0.332 | 0.321 | 0.339 | 0.320 | 0.346 | 0.332 | 0.346 | 0.381 | 0.371 | 0.387 | 0.393 | 0.390 | 0.396 | 0.441 | 0.405 | 0.437 | 0.448 | 0.377 | 0.382 | 0.739 | 0.558 | 0.611 | 0.500 |
| | Avg | 0.245 | 0.282 | 0.234 | 0.273 | 0.238 | 0.275 | 0.241 | 0.283 | 0.242 | 0.279 | 0.279 | 0.301 | 0.284 | 0.324 | 0.300 | 0.342 | 0.318 | 0.323 | 0.318 | 0.360 | 0.289 | 0.322 | 0.597 | 0.495 | 0.546 | 0.469 |
| Electricity | 96 | 0.160 | 0.269 | 0.139 | 0.241 | 0.139 | 0.237 | 0.150 | 0.253 | 0.140 | 0.238 | 0.299 | 0.373 | 0.231 | 0.323 | 0.261 | 0.348 | 0.420 | 0.466 | 0.599 | 0.587 | 0.350 | 0.425 | 1.259 | 0.919 | 0.993 | 0.784 |
| | 192 | 0.174 | 0.279 | 0.151 | 0.248 | 0.156 | 0.252 | 0.164 | 0.264 | 0.160 | 0.255 | 0.305 | 0.379 | 0.261 | 0.356 | 0.338 | 0.406 | 0.411 | 0.459 | 0.620 | 0.598 | 0.376 | 0.448 | 1.160 | 0.873 | 0.938 | 0.753 |
| | 336 | 0.190 | 0.294 | 0.169 | 0.270 | 0.175 | 0.270 | 0.181 | 0.282 | 0.180 | 0.276 | 0.319 | 0.391 | 0.360 | 0.445 | 0.410 | 0.474 | 0.434 | 0.473 | 0.662 | 0.619 | 0.428 | 0.485 | 1.157 | 0.872 | 0.925 | 0.745 |
| | 720 | 0.229 | 0.323 | 0.240 | 0.322 | 0.233 | 0.317 | 0.223 | 0.321 | 0.241 | 0.323 | 0.369 | 0.426 | 0.530 | 0.585 | 0.715 | 0.685 | 0.510 | 0.521 | 0.757 | 0.664 | 0.611 | 0.597 | 1.203 | 0.898 | 1.004 | 0.790 |
| | Avg | 0.198 | 0.291 | 0.175 | 0.270 | 0.176 | 0.269 | 0.180 | 0.280 | 0.180 | 0.273 | 0.323 | 0.392 | 0.346 | 0.427 | 0.431 | 0.478 | 0.444 | 0.480 | 0.660 | 0.617 | 0.441 | 0.489 | 1.195 | 0.891 | 0.965 | 0.768 |
| Traffic | 96 | 0.465 | 0.349 | 0.418 | 0.291 | 0.414 | 0.297 | 0.419 | 0.298 | 0.403 | 0.289 | 0.719 | 0.416 | 0.639 | 0.400 | 0.672 | 0.405 | 1.412 | 0.802 | 1.643 | 0.855 | 1.157 | 0.636 | 1.557 | 0.821 | 1.527 | 0.815 |
| | 192 | 0.468 | 0.350 | 0.414 | 0.296 | 0.426 | 0.301 | 0.434 | 0.305 | 0.415 | 0.296 | 0.748 | 0.428 | 0.637 | 0.416 | 0.727 | 0.424 | 1.419 | 0.806 | 1.641 | 0.854 | 1.207 | 0.661 | 1.454 | 0.765 | 1.538 | 0.817 |
| | 336 | 0.483 | 0.356 | 0.421 | 0.311 | 0.434 | 0.303 | 0.449 | 0.313 | 0.426 | 0.304 | 0.853 | 0.471 | 0.655 | 0.427 | 0.749 | 0.454 | 1.443 | 0.815 | 1.711 | 0.878 | 1.334 | 0.713 | 1.521 | 0.812 | 1.550 | 0.819 |
| | 720 | 0.520 | 0.373 | 0.462 | 0.327 | 0.487 | 0.337 | 0.484 | 0.336 | 0.474 | 0.331 | 1.485 | 0.825 | 0.722 | 0.456 | 0.847 | 0.499 | 1.539 | 0.837 | 2.660 | 1.157 | 1.292 | 0.726 | 1.605 | 0.846 | 1.588 | 0.833 |
| | Avg | 0.484 | 0.357 | 0.429 | 0.306 | 0.440 | 0.310 | 0.447 | 0.313 | 0.430 | 0.305 | 0.951 | 0.535 | 0.663 | 0.425 | 0.749 | 0.446 | 1.453 | 0.815 | 1.914 | 0.936 | 1.248 | 0.684 | 1.534 | 0.811 | 1.551 | 0.821 |

## B.2. Zero-shot Forecasting

Table 14: Full zero-shot learning results on ETT datasets. A lower value indicates better performance.

| Methods | | Time-VLM | | Time-LLM | | LLMTime | | GPT4TS | | DLinear | | PatchTST | | TimesNet | | Autoformer | |
|---|---|---|---|---|---|---|---|---|---|---|---|---|---|---|---|---|---|
| Metric | | MSE | MAE | MSE | MAE | MSE | MAE | MSE | MAE | MSE | MAE | MSE | MAE | MSE | MAE | MSE | MAE |
| ETTh1→ETTh2 | 96 | **0.277** | **0.338** | 0.279 | 0.337 | 0.510 | 0.576 | 0.335 | 0.374 | 0.347 | 0.400 | 0.304 | 0.350 | 0.358 | 0.387 | 0.469 | 0.486 |
| | 192 | **0.333** | **0.378** | 0.351 | 0.374 | 0.523 | 0.586 | 0.412 | 0.417 | 0.447 | 0.460 | 0.386 | 0.400 | 0.427 | 0.429 | 0.634 | 0.567 |
| | 336 | **0.360** | **0.399** | 0.388 | 0.415 | 0.640 | 0.637 | 0.441 | 0.444 | 0.515 | 0.505 | 0.414 | 0.428 | 0.449 | 0.451 | 0.655 | 0.588 |
| | 720 | **0.383** | **0.425** | 0.391 | 0.420 | 2.296 | 1.034 | 0.438 | 0.452 | 0.665 | 0.589 | 0.419 | 0.443 | 0.448 | 0.458 | 0.570 | 0.549 |
| | Avg | **0.338** | **0.385** | 0.353 | 0.387 | 0.992 | 0.708 | 0.406 | 0.422 | 0.493 | 0.488 | 0.380 | 0.405 | 0.421 | 0.431 | 0.582 | 0.548 |
| ETTh1→ETTm2 | 96 | 0.207 | 0.297 | **0.189** | **0.293** | 0.646 | 0.563 | 0.236 | 0.315 | 0.255 | 0.357 | 0.215 | 0.304 | 0.239 | 0.313 | 0.352 | 0.432 |
| | 192 | 0.258 | 0.329 | **0.237** | **0.312** | 0.934 | 0.654 | 0.287 | 0.342 | 0.338 | 0.413 | 0.275 | 0.339 | 0.291 | 0.342 | 0.413 | 0.460 |
| | 336 | 0.310 | 0.360 | **0.291** | **0.365** | 1.157 | 0.728 | 0.341 | 0.374 | 0.425 | 0.465 | 0.334 | 0.373 | 0.342 | 0.371 | 0.465 | 0.489 |
| | 720 | 0.398 | 0.412 | **0.372** | **0.390** | 4.730 | 1.531 | 0.435 | 0.422 | 0.640 | 0.573 | 0.431 | 0.424 | 0.434 | 0.419 | 0.599 | 0.551 |
| | Avg | 0.293 | 0.350 | **0.273** | **0.340** | 1.867 | 0.869 | 0.325 | 0.363 | 0.415 | 0.452 | 0.314 | 0.360 | 0.327 | 0.361 | 0.457 | 0.483 |
| ETTh2→ETTh1 | 96 | **0.434** | **0.441** | 0.450 | 0.452 | 1.130 | 0.777 | 0.732 | 0.577 | 0.689 | 0.555 | 0.485 | 0.465 | 0.848 | 0.601 | 0.693 | 0.569 |
| | 192 | **0.464** | **0.454** | 0.465 | 0.461 | 1.242 | 0.820 | 0.758 | 0.559 | 0.707 | 0.568 | 0.565 | 0.509 | 0.860 | 0.610 | 0.760 | 0.601 |
| | 336 | **0.489** | **0.481** | 0.501 | 0.482 | 1.328 | 0.864 | 0.759 | 0.578 | 0.710 | 0.577 | 0.581 | 0.515 | 0.867 | 0.626 | 0.781 | 0.619 |
| | 720 | 0.595 | 0.543 | **0.501** | **0.502** | 4.145 | 1.461 | 0.781 | 0.597 | 0.704 | 0.596 | 0.628 | 0.561 | 0.887 | 0.648 | 0.796 | 0.644 |
| | Avg | 0.496 | 0.480 | **0.479** | **0.474** | 1.961 | 0.981 | 0.757 | 0.578 | 0.703 | 0.574 | 0.565 | 0.513 | 0.865 | 0.621 | 0.757 | 0.608 |
| ETTh2→ETTm2 | 96 | 0.204 | 0.297 | **0.174** | **0.276** | 0.646 | 0.563 | 0.253 | 0.329 | 0.240 | 0.336 | 0.226 | 0.309 | 0.248 | 0.324 | 0.263 | 0.352 |
| | 192 | 0.255 | 0.328 | **0.233** | **0.315** | 0.934 | 0.654 | 0.293 | 0.346 | 0.295 | 0.369 | 0.289 | 0.345 | 0.296 | 0.352 | 0.326 | 0.389 |
| | 336 | 0.311 | 0.362 | **0.291** | **0.337** | 1.157 | 0.728 | 0.347 | 0.376 | 0.345 | 0.397 | 0.348 | 0.379 | 0.353 | 0.383 | 0.387 | 0.426 |
| | 720 | 0.420 | 0.425 | **0.392** | **0.417** | 4.730 | 1.531 | 0.446 | 0.429 | 0.432 | 0.442 | 0.439 | 0.427 | 0.471 | 0.446 | 0.487 | 0.478 |
| | Avg | 0.297 | 0.353 | **0.272** | **0.341** | 1.867 | 0.869 | 0.335 | 0.370 | 0.328 | 0.386 | 0.325 | 0.365 | 0.342 | 0.376 | 0.366 | 0.411 |
| ETTm1→ETTh2 | 96 | **0.297** | **0.356** | 0.321 | 0.369 | 0.510 | 0.576 | 0.353 | 0.392 | 0.365 | 0.415 | 0.354 | 0.385 | 0.377 | 0.407 | 0.435 | 0.470 |
| | 192 | **0.349** | **0.388** | 0.389 | 0.410 | 0.523 | 0.586 | 0.443 | 0.437 | 0.454 | 0.462 | 0.447 | 0.434 | 0.471 | 0.453 | 0.495 | 0.489 |
| | 336 | **0.374** | **0.409** | 0.408 | 0.433 | 0.640 | 0.637 | 0.469 | 0.461 | 0.496 | 0.494 | 0.481 | 0.463 | 0.472 | 0.484 | 0.470 | 0.472 |
| | 720 | **0.396** | **0.433** | 0.406 | 0.436 | 2.296 | 1.034 | 0.466 | 0.468 | 0.541 | 0.529 | 0.474 | 0.471 | 0.495 | 0.482 | 0.480 | 0.485 |
| | Avg | **0.354** | **0.397** | 0.381 | 0.412 | 0.992 | 0.708 | 0.433 | 0.439 | 0.464 | 0.475 | 0.439 | 0.438 | 0.457 | 0.454 | 0.470 | 0.479 |
| ETTm1→ETTm2 | 96 | 0.178 | 0.264 | **0.169** | **0.257** | 0.646 | 0.563 | 0.217 | 0.294 | 0.221 | 0.314 | 0.195 | 0.271 | 0.222 | 0.295 | 0.385 | 0.457 |
| | 192 | **0.226** | **0.298** | 0.227 | 0.318 | 0.934 | 0.654 | 0.277 | 0.327 | 0.286 | 0.359 | 0.258 | 0.311 | 0.288 | 0.337 | 0.433 | 0.469 |
| | 336 | **0.279** | **0.329** | 0.290 | 0.338 | 1.157 | 0.728 | 0.331 | 0.360 | 0.357 | 0.406 | 0.317 | 0.348 | 0.341 | 0.367 | 0.476 | 0.477 |
| | 720 | **0.373** | **0.385** | 0.375 | **0.367** | 4.730 | 1.531 | 0.429 | 0.413 | 0.476 | 0.476 | 0.416 | 0.404 | 0.436 | 0.418 | 0.582 | 0.535 |
| | Avg | **0.264** | **0.319** | 0.268 | 0.320 | 1.867 | 0.869 | 0.313 | 0.348 | 0.335 | 0.389 | 0.296 | 0.334 | 0.322 | 0.354 | 0.469 | 0.484 |
| ETTm2→ETTh2 | 96 | **0.285** | **0.347** | 0.298 | 0.356 | 0.510 | 0.576 | 0.360 | 0.401 | 0.333 | 0.391 | 0.327 | 0.367 | 0.360 | 0.401 | 0.353 | 0.393 |
| | 192 | **0.348** | **0.384** | 0.359 | 0.397 | 0.523 | 0.586 | 0.434 | 0.437 | 0.441 | 0.456 | 0.411 | 0.418 | 0.434 | 0.437 | 0.432 | 0.437 |
| | 336 | 0.380 | 0.415 | **0.367** | **0.412** | 0.640 | 0.637 | 0.460 | 0.459 | 0.505 | 0.503 | 0.439 | 0.447 | 0.460 | 0.459 | 0.452 | 0.459 |
| | 720 | 0.424 | 0.451 | **0.393** | **0.434** | 2.296 | 1.034 | 0.485 | 0.477 | 0.543 | 0.534 | 0.459 | 0.470 | 0.485 | 0.477 | 0.453 | 0.467 |
| | Avg | 0.359 | **0.399** | **0.354** | 0.400 | 0.992 | 0.708 | 0.435 | 0.443 | 0.455 | 0.471 | 0.409 | 0.425 | 0.435 | 0.443 | 0.423 | 0.439 |
| ETTm2→ETTm1 | 96 | 0.370 | 0.390 | **0.359** | **0.397** | 1.179 | 0.781 | 0.747 | 0.558 | 0.570 | 0.490 | 0.491 | 0.437 | 0.747 | 0.558 | 0.735 | 0.576 |
| | 192 | 0.400 | **0.409** | **0.390** | 0.420 | 1.327 | 0.846 | 0.781 | 0.560 | 0.590 | 0.506 | 0.530 | 0.470 | 0.781 | 0.560 | 0.753 | 0.586 |
| | 336 | 0.426 | 0.420 | **0.421** | **0.445** | 1.478 | 0.902 | 0.778 | 0.578 | 0.706 | 0.567 | 0.565 | 0.497 | 0.778 | 0.578 | 0.750 | 0.593 |
| | 720 | 0.531 | 0.487 | **0.487** | 0.488 | 3.749 | 1.408 | 0.769 | 0.573 | 0.731 | 0.584 | 0.686 | 0.565 | 0.769 | 0.573 | 0.782 | 0.609 |
| | Avg | 0.432 | **0.426** | **0.414** | 0.438 | 1.933 | 0.984 | 0.769 | 0.567 | 0.649 | 0.537 | 0.568 | 0.492 | 0.769 | 0.567 | 0.755 | 0.591 |

## B.3. Short-term Forecasting

## B.4. Long-term Forecasting

# C. Visualizations

## C.1. Visualization of Generated Time Series Images

The image generation module employs advanced techniques—frequency and periodicity Encoding, multi-scale convolution, interpolation and normalization—to create informative and discriminative image representations of time series data. These representations enhance downstream VLMs for improved forecasting. As shown in Figure 6, the generated images capture key temporal characteristics through the following features:

- **Frequency-Domain Information**: FFT integration captures frequency-domain characteristics, visualized as distinct textures—fine-grained for high-frequency components and broader color regions for low-frequency components.
- **Multi-scale Periodic Encoding**: Temporal dependencies at multiple scales (e.g., daily, weekly) are encoded, visible as regular patterns such as repeating vertical bands for daily cycles or broader horizontal patterns for weekly cycles.
- **Image Interpolation**: Bilinear interpolation ensures smooth and coherent images, preserving essential time series characteristics through seamless transitions between color intensities.

Table 15: Full short-term time series forecasting results. The forecasting horizons are in [6, 48] and the last three rows are weighted averaged from all datasets under different sampling intervals. A lower value indicates better performance.

| | Methods | Time-VLM | Time-LLM | GPT4TS | TimesNet | PatchTST | N-HiTS | N-BEATS | ETSformer | LightTS | DLinear | FEDformer | Stationary | Autoformer | Informer | Reformer |
|---|---|---|---|---|---|---|---|---|---|---|---|---|---|---|---|---|
| Yearly | SMAPE | **13.285** | 13.419 | 15.110 | 15.378 | 13.477 | 13.422 | 13.487 | 18.009 | 14.247 | 16.965 | 14.021 | 13.717 | 13.974 | 14.727 | 16.169 |
| | MASE | **2.993** | 3.005 | 3.565 | 3.554 | 3.019 | 3.056 | 3.036 | 4.487 | 3.109 | 4.283 | 3.036 | 3.078 | 3.134 | 3.418 | 3.800 |
| | OWA | **0.783** | 0.789 | 0.911 | 0.918 | 0.792 | 0.795 | 0.795 | 1.115 | 0.827 | 1.058 | 0.811 | 0.807 | 0.822 | 0.881 | 0.973 |
| Quarterly | SMAPE | 10.218 | **10.110** | 10.597 | 10.465 | 10.380 | 10.185 | 10.564 | 13.376 | 11.364 | 12.145 | 11.100 | 10.958 | 11.338 | 11.360 | 13.313 |
| | MASE | 1.203 | **1.178** | 1.253 | 1.227 | 1.233 | 1.180 | 1.252 | 1.906 | 1.328 | 1.520 | 1.350 | 1.325 | 1.365 | 1.401 | 1.775 |
| | OWA | 0.903 | **0.889** | 0.938 | 0.923 | 0.921 | 0.893 | 0.936 | 1.302 | 1.000 | 1.106 | 0.996 | 0.981 | 1.012 | 1.027 | 1.252 |
| Monthly | SMAPE | **12.788** | 12.980 | 13.258 | 13.513 | 12.959 | 13.059 | 13.089 | 14.588 | 14.014 | 13.514 | 14.403 | 13.917 | 13.958 | 14.062 | 20.128 |
| | MASE | **0.942** | 0.963 | 1.003 | 1.039 | 0.970 | 1.013 | 0.996 | 1.368 | 1.053 | 1.037 | 1.147 | 1.097 | 1.103 | 1.141 | 2.614 |
| | OWA | **0.886** | 0.903 | 0.931 | 0.957 | 0.905 | 0.929 | 0.922 | 1.149 | 0.981 | 0.956 | 1.038 | 0.998 | 1.002 | 1.024 | 1.927 |
| Others | SMAPE | 4.945 | **4.795** | 6.124 | 6.913 | 4.952 | 4.711 | 6.599 | 7.267 | 15.880 | 6.709 | 7.148 | 6.302 | 5.485 | 24.460 | 32.491 |
| | MASE | 3.257 | **3.178** | 4.116 | 4.507 | 3.347 | 3.054 | 4.430 | 5.240 | 11.434 | 4.953 | 4.041 | 4.064 | 3.865 | 20.960 | 33.355 |
| | OWA | 1.034 | **1.006** | 1.259 | 1.438 | 1.049 | 0.977 | 1.393 | 1.591 | 3.474 | 1.487 | 1.389 | 1.304 | 1.187 | 5.879 | 8.679 |
| Average | SMAPE | **11.894** | 11.983 | 12.690 | 12.880 | 12.059 | 12.035 | 12.250 | 14.718 | 13.525 | 13.639 | 13.160 | 12.780 | 12.909 | 14.086 | 18.200 |
| | MASE | **1.592** | 1.595 | 1.808 | 1.836 | 1.623 | 1.625 | 1.698 | 2.408 | 2.111 | 2.095 | 1.775 | 1.756 | 1.771 | 2.718 | 4.223 |
| | OWA | **0.855** | 0.859 | 0.940 | 0.955 | 0.869 | 0.869 | 0.896 | 1.172 | 1.051 | 1.051 | 0.949 | 0.930 | 0.939 | 1.230 | 1.775 |

Table 16: Full long-term forecasting results. We use the same protocol as in Table 1.

| Methods | | Time-VLM | | Time-LLM | | GPT4TS | | DLinear | | PatchTST | | TimesNet | | FEDformer | | Autoformer | | Stationary | | ETSformer | | LightTS | | Informer | | Reformer | |
|---|---|---|---|---|---|---|---|---|---|---|---|---|---|---|---|---|---|---|---|---|---|---|---|---|---|---|---|
| Metric | | MSE | MAE | MSE | MAE | MSE | MAE | MSE | MAE | MSE | MAE | MSE | MAE | MSE | MAE | MSE | MAE | MSE | MAE | MSE | MAE | MSE | MAE | MSE | MAE | MSE | MAE |
| ETTh1 | 96 | **0.361** | **0.386** | 0.362 | 0.392 | 0.376 | 0.397 | 0.375 | 0.399 | 0.370 | 0.399 | 0.384 | 0.402 | 0.376 | 0.419 | 0.449 | 0.459 | 0.513 | 0.491 | 0.494 | 0.479 | 0.424 | 0.432 | 0.865 | 0.713 | 0.837 | 0.728 |
| | 192 | **0.397** | **0.415** | 0.398 | 0.418 | 0.416 | 0.418 | 0.405 | 0.416 | 0.413 | 0.421 | 0.436 | 0.429 | 0.420 | 0.448 | 0.500 | 0.482 | 0.534 | 0.504 | 0.538 | 0.504 | 0.475 | 0.462 | 1.008 | 0.792 | 0.923 | 0.766 |
| | 336 | **0.420** | **0.421** | 0.430 | 0.427 | 0.442 | 0.433 | 0.439 | 0.443 | 0.422 | 0.436 | 0.491 | 0.469 | 0.459 | 0.465 | 0.521 | 0.496 | 0.588 | 0.535 | 0.574 | 0.521 | 0.518 | 0.488 | 1.107 | 0.809 | 1.097 | 0.835 |
| | 720 | **0.441** | **0.458** | 0.442 | 0.457 | 0.477 | 0.456 | 0.472 | 0.490 | 0.447 | 0.466 | 0.521 | 0.500 | 0.506 | 0.507 | 0.514 | 0.512 | 0.643 | 0.616 | 0.562 | 0.535 | 0.547 | 0.533 | 1.181 | 0.865 | 1.257 | 0.889 |
| | Avg | **0.405** | **0.420** | 0.408 | 0.423 | 0.465 | 0.455 | 0.422 | 0.437 | 0.413 | 0.430 | 0.458 | 0.450 | 0.440 | 0.460 | 0.496 | 0.487 | 0.570 | 0.537 | 0.542 | 0.510 | 0.491 | 0.479 | 1.040 | 0.795 | 1.029 | 0.805 |
| ETTh2 | 96 | **0.267** | 0.335 | 0.268 | 0.328 | 0.285 | 0.342 | 0.289 | 0.353 | 0.274 | 0.336 | 0.340 | 0.374 | 0.358 | 0.397 | 0.346 | 0.388 | 0.476 | 0.458 | 0.340 | 0.391 | 0.397 | 0.437 | 3.755 | 1.525 | 2.626 | 1.317 |
| | 192 | **0.326** | **0.373** | 0.329 | 0.375 | 0.354 | 0.389 | 0.383 | 0.418 | 0.339 | 0.379 | 0.402 | 0.414 | 0.429 | 0.439 | 0.456 | 0.452 | 0.512 | 0.493 | 0.430 | 0.439 | 0.520 | 0.504 | 5.602 | 1.931 | 11.120 | 2.979 |
| | 336 | 0.357 | 0.406 | 0.368 | 0.409 | 0.373 | 0.407 | 0.448 | 0.465 | **0.329** | **0.380** | 0.452 | 0.452 | 0.496 | 0.487 | 0.482 | 0.486 | 0.552 | 0.551 | 0.485 | 0.479 | 0.626 | 0.559 | 4.721 | 1.835 | 9.323 | 2.769 |
| | 720 | 0.412 | 0.449 | 0.372 | 0.420 | 0.406 | 0.441 | 0.605 | 0.551 | 0.379 | 0.422 | 0.462 | 0.468 | 0.463 | 0.474 | 0.515 | 0.511 | 0.562 | 0.560 | 0.500 | 0.497 | 0.863 | 0.672 | 3.647 | 1.625 | 3.874 | 1.697 |
| | Avg | 0.341 | 0.391 | 0.334 | 0.383 | 0.381 | 0.412 | 0.431 | 0.446 | **0.330** | **0.379** | 0.414 | 0.427 | 0.437 | 0.449 | 0.450 | 0.459 | 0.526 | 0.516 | 0.439 | 0.452 | 0.602 | 0.543 | 4.431 | 1.729 | 6.736 | 2.191 |
| ETTm1 | 96 | 0.304 | 0.346 | 0.272 | **0.334** | 0.292 | 0.346 | 0.299 | 0.343 | 0.290 | 0.342 | 0.338 | 0.375 | 0.379 | 0.419 | 0.505 | 0.475 | 0.386 | 0.398 | 0.375 | 0.398 | 0.374 | 0.400 | 0.672 | 0.571 | 0.538 | 0.528 |
| | 192 | 0.332 | 0.366 | 0.310 | 0.358 | 0.332 | 0.372 | 0.335 | 0.365 | 0.332 | 0.369 | 0.374 | 0.387 | 0.426 | 0.441 | 0.553 | 0.496 | 0.459 | 0.444 | 0.408 | 0.410 | 0.400 | 0.407 | 0.795 | 0.669 | 0.658 | 0.592 |
| | 336 | 0.364 | **0.383** | 0.352 | 0.384 | 0.366 | 0.394 | 0.369 | 0.386 | 0.366 | 0.392 | 0.410 | 0.411 | 0.445 | 0.459 | 0.621 | 0.537 | 0.495 | 0.464 | 0.435 | 0.428 | 0.438 | 0.438 | 1.212 | 0.871 | 0.898 | 0.721 |
| | 720 | 0.402 | **0.410** | 0.383 | 0.411 | 0.417 | 0.421 | 0.425 | 0.421 | 0.416 | 0.420 | 0.478 | 0.450 | 0.543 | 0.490 | 0.671 | 0.561 | 0.585 | 0.516 | 0.499 | 0.462 | 0.527 | 0.502 | 1.166 | 0.823 | 1.102 | 0.841 |
| | Avg | 0.350 | 0.377 | 0.329 | 0.372 | 0.388 | 0.403 | 0.357 | 0.378 | 0.351 | 0.380 | 0.400 | 0.406 | 0.448 | 0.452 | 0.588 | 0.517 | 0.481 | 0.456 | 0.429 | 0.425 | 0.435 | 0.437 | 0.961 | 0.734 | 0.799 | 0.671 |
| ETTm2 | 96 | **0.160** | **0.250** | 0.161 | 0.253 | 0.173 | 0.262 | 0.167 | 0.260 | 0.165 | 0.255 | 0.187 | 0.267 | 0.203 | 0.287 | 0.255 | 0.339 | 0.192 | 0.274 | 0.189 | 0.280 | 0.209 | 0.308 | 0.365 | 0.453 | 0.658 | 0.619 |
| | 192 | **0.215** | **0.291** | 0.219 | 0.293 | 0.229 | 0.301 | 0.224 | 0.303 | 0.220 | 0.292 | 0.249 | 0.309 | 0.269 | 0.328 | 0.281 | 0.340 | 0.280 | 0.339 | 0.253 | 0.319 | 0.311 | 0.382 | 0.533 | 0.563 | 1.078 | 0.827 |
| | 336 | **0.270** | **0.325** | 0.271 | 0.329 | 0.286 | 0.341 | 0.281 | 0.342 | 0.274 | 0.329 | 0.321 | 0.351 | 0.325 | 0.366 | 0.339 | 0.372 | 0.334 | 0.361 | 0.314 | 0.357 | 0.442 | 0.466 | 1.363 | 0.887 | 1.549 | 0.972 |
| | 720 | **0.348** | **0.378** | 0.352 | 0.379 | 0.378 | 0.401 | 0.397 | 0.421 | 0.362 | 0.385 | 0.408 | 0.403 | 0.421 | 0.415 | 0.433 | 0.432 | 0.417 | 0.413 | 0.414 | 0.413 | 0.675 | 0.587 | 3.379 | 1.338 | 2.631 | 1.242 |
| | Avg | **0.248** | **0.311** | 0.251 | 0.313 | 0.284 | 0.339 | 0.267 | 0.333 | 0.255 | 0.315 | 0.291 | 0.333 | 0.305 | 0.349 | 0.327 | 0.371 | 0.306 | 0.347 | 0.293 | 0.342 | 0.409 | 0.436 | 1.410 | 0.810 | 1.479 | 0.915 |
| Weather | 96 | 0.148 | 0.200 | 0.147 | 0.201 | 0.162 | 0.212 | 0.176 | 0.237 | 0.149 | **0.198** | 0.172 | 0.220 | 0.217 | 0.296 | 0.266 | 0.336 | 0.173 | 0.223 | 0.197 | 0.281 | 0.182 | 0.242 | 0.300 | 0.384 | 0.689 | 0.596 |
| | 192 | 0.193 | 0.240 | 0.189 | 0.234 | 0.204 | 0.248 | 0.220 | 0.282 | 0.194 | 0.241 | 0.219 | 0.261 | 0.276 | 0.336 | 0.307 | 0.367 | 0.245 | 0.285 | 0.237 | 0.312 | 0.227 | 0.287 | 0.598 | 0.544 | 0.752 | 0.638 |
| | 336 | 0.243 | 0.281 | 0.262 | 0.279 | 0.254 | 0.286 | 0.265 | 0.319 | 0.245 | 0.282 | 0.280 | 0.306 | 0.339 | 0.380 | 0.359 | 0.395 | 0.321 | 0.338 | 0.298 | 0.353 | 0.282 | 0.334 | 0.578 | 0.523 | 0.639 | 0.596 |
| | 720 | 0.312 | 0.332 | 0.304 | 0.316 | 0.326 | 0.337 | 0.333 | 0.362 | 0.314 | 0.334 | 0.365 | 0.359 | 0.403 | 0.428 | 0.419 | 0.428 | 0.414 | 0.410 | 0.352 | 0.288 | 0.352 | 0.386 | 1.059 | 0.741 | 1.130 | 0.792 |
| | Avg | 0.224 | 0.263 | 0.225 | 0.257 | 0.237 | 0.270 | 0.248 | 0.300 | 0.225 | 0.264 | 0.259 | 0.287 | 0.309 | 0.360 | 0.338 | 0.382 | 0.288 | 0.314 | 0.271 | 0.334 | 0.261 | 0.312 | 0.634 | 0.548 | 0.803 | 0.656 |
| Electricity | 96 | 0.142 | 0.245 | 0.131 | 0.224 | 0.139 | 0.238 | 0.140 | 0.237 | **0.129** | **0.222** | 0.168 | 0.272 | 0.193 | 0.308 | 0.201 | 0.317 | 0.169 | 0.273 | 0.187 | 0.304 | 0.207 | 0.307 | 0.274 | 0.368 | 0.312 | 0.402 |
| | 192 | 0.157 | 0.260 | 0.152 | 0.241 | 0.153 | 0.251 | 0.153 | 0.249 | 0.157 | 0.240 | 0.184 | 0.289 | 0.201 | 0.315 | 0.222 | 0.334 | 0.182 | 0.286 | 0.199 | 0.315 | 0.213 | 0.316 | 0.296 | 0.386 | 0.348 | 0.433 |
| | 336 | 0.174 | 0.276 | 0.160 | 0.248 | 0.169 | 0.266 | 0.169 | 0.267 | 0.163 | 0.259 | 0.198 | 0.300 | 0.214 | 0.329 | 0.231 | 0.338 | 0.200 | 0.304 | 0.212 | 0.329 | 0.230 | 0.333 | 0.300 | 0.394 | 0.350 | 0.433 |
| | 720 | 0.214 | 0.308 | 0.192 | 0.298 | 0.206 | 0.297 | 0.203 | 0.301 | 0.197 | 0.290 | 0.220 | 0.320 | 0.246 | 0.355 | 0.254 | 0.361 | 0.222 | 0.321 | 0.233 | 0.345 | 0.265 | 0.360 | 0.373 | 0.439 | 0.340 | 0.420 |
| | Avg | 0.172 | 0.273 | 0.158 | 0.252 | 0.167 | 0.263 | 0.166 | 0.263 | 0.161 | 0.252 | 0.192 | 0.295 | 0.214 | 0.327 | 0.227 | 0.338 | 0.193 | 0.296 | 0.208 | 0.323 | 0.229 | 0.329 | 0.311 | 0.397 | 0.338 | 0.422 |
| Traffic | 96 | 0.393 | 0.290 | 0.362 | 0.248 | 0.388 | 0.282 | 0.410 | 0.282 | 0.360 | 0.249 | 0.593 | 0.321 | 0.587 | 0.366 | 0.613 | 0.388 | 0.612 | 0.338 | 0.607 | 0.392 | 0.615 | 0.391 | 0.719 | 0.391 | 0.732 | 0.423 |
| | 192 | 0.405 | 0.296 | 0.374 | 0.247 | 0.407 | 0.290 | 0.423 | 0.287 | 0.379 | 0.256 | 0.617 | 0.336 | 0.604 | 0.373 | 0.616 | 0.382 | 0.613 | 0.340 | 0.621 | 0.399 | 0.601 | 0.382 | 0.696 | 0.379 | 0.733 | 0.420 |
| | 336 | 0.420 | 0.305 | 0.385 | 0.271 | 0.412 | 0.294 | 0.436 | 0.296 | 0.392 | 0.264 | 0.629 | 0.336 | 0.621 | 0.383 | 0.622 | 0.337 | 0.618 | 0.328 | 0.622 | 0.396 | 0.613 | 0.386 | 0.777 | 0.420 | 0.742 | 0.420 |
| | 720 | 0.459 | 0.323 | 0.430 | 0.288 | 0.450 | 0.312 | 0.466 | 0.315 | 0.432 | 0.286 | 0.640 | 0.350 | 0.626 | 0.382 | 0.660 | 0.408 | 0.653 | 0.355 | 0.632 | 0.396 | 0.658 | 0.407 | 0.864 | 0.472 | 0.755 | 0.423 |
| | Avg | 0.419 | 0.303 | 0.388 | 0.264 | 0.414 | 0.294 | 0.433 | 0.295 | 0.390 | 0.263 | 0.620 | 0.336 | 0.610 | 0.376 | 0.628 | 0.379 | 0.624 | 0.340 | 0.621 | 0.396 | 0.622 | 0.392 | 0.764 | 0.416 | 0.741 | 0.422 |

- **Color Trends**: Color intensity corresponds to time series values—darker regions (e.g., deep blue) indicate lower values, while brighter regions (e.g., yellow) represent higher values, enabling easy identification of trends.

- **Abrupt Changes and Anomalies**: Sudden shifts in color intensity (e.g., sharp transitions from dark to bright) highlight abrupt changes or anomalies, crucial for identifying irregular events like traffic spikes or weather shifts.

## C.2. Visualization of prediction results

The prediction results in Figures 7, 8, 9, and 10 demonstrate Time-VLM's ability to accurately forecast time series across diverse datasets and prediction horizons. For datasets with clear periodic structures, such as the daily cycles in ETTh1 and ETTm1, Time-VLM captures both global trends and fine-grained temporal patterns effectively. This is evident in the close

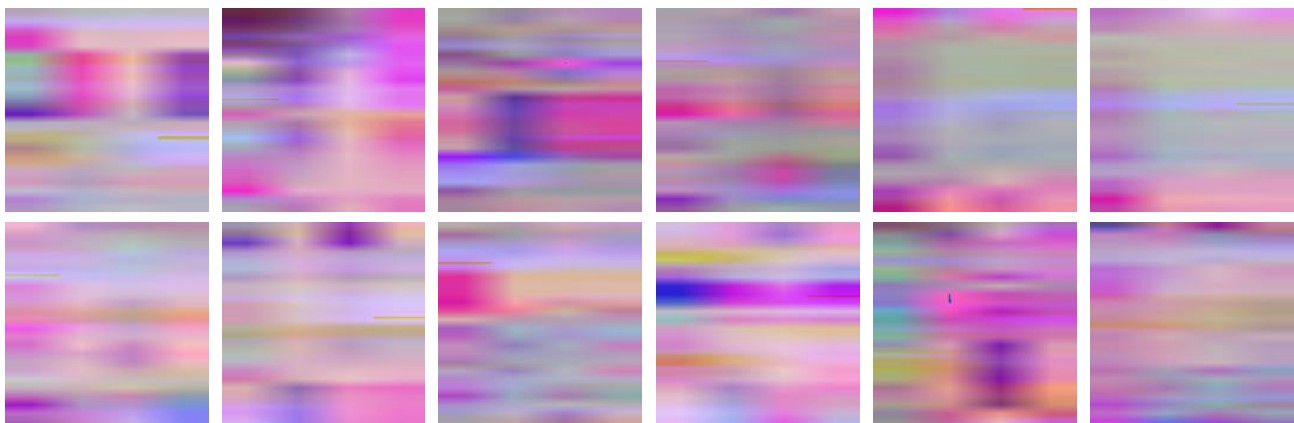

Figure 6: Time series transformed images, capturing key temporal characteristics, including trends, stationarity, seasonality, sudden changes, and frequency-domain patterns.

alignment between the true values (solid lines) and predicted values (dashed lines) across all horizons. Similarly, for the ECL dataset, which exhibits regular consumption patterns, Time-VLM delivers highly accurate forecasts, showcasing its strength in handling structured environments.

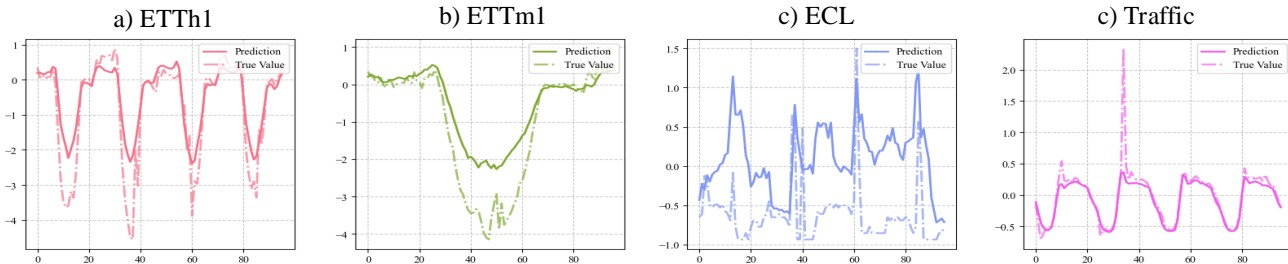

Figure 7: Prediction results visualization for ETTh1, ETTm1, ECL, and Traffic datasets at 96 prediction lengths. True values (solid line) and predicted values (dashed line) are shown for each dataset and horizon.

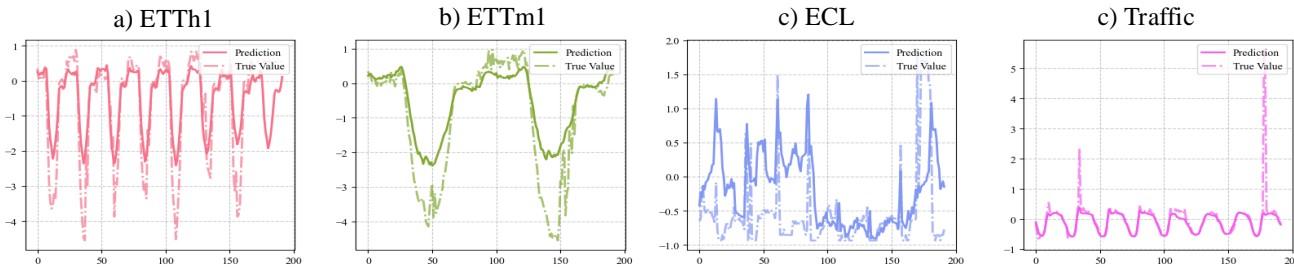

Figure 8: Prediction results visualization for ETTh1, ETTm1, ECL, and Traffic datasets at 192 prediction lengths. True values (solid line) and predicted values (dashed line) are shown for each dataset and horizon.

However, performance varies for datasets with irregular or abrupt changes. On the Traffic dataset, which is characterized by non-stationary patterns, Time-VLM shows slight deviations in capturing sudden fluctuations, particularly at longer horizons (e.g., 336 and 720). These deviations highlight the challenges of modeling highly irregular data and suggest opportunities for refining the time series-to-image transformation process to better handle such scenarios.

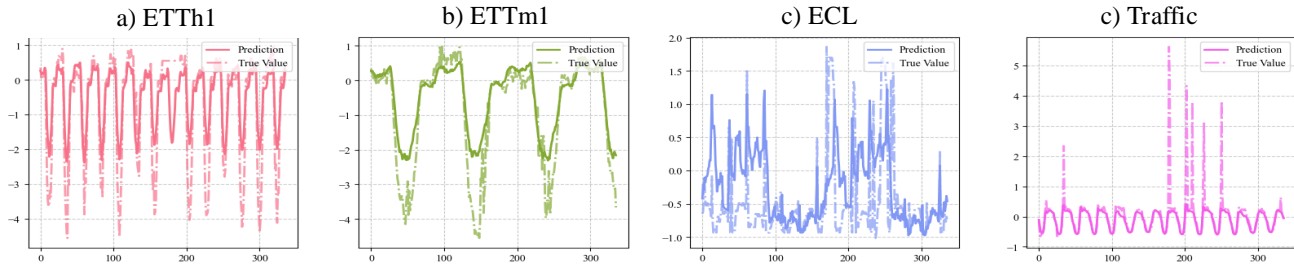

Figure 9: Prediction results visualization for ETTh1, ETTm1, ECL, and Traffic datasets at 336 prediction lengths. True values (solid line) and predicted values (dashed line) are shown for each dataset and horizon.

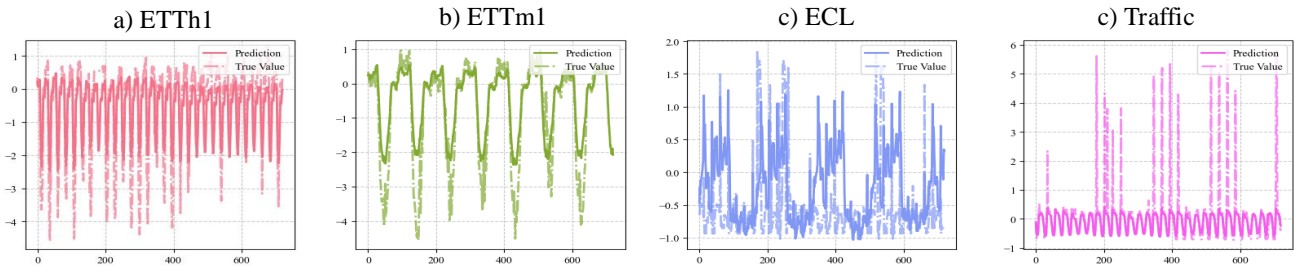

Figure 10: Prediction results visualization for ETTh1, ETTm1, ECL, and Traffic datasets at 720 prediction lengths. True values (solid line) and predicted values (dashed line) are shown for each dataset and horizon.

# D. Future Work

### D.1. Limitations

While Time-VLM demonstrates significant improvements in time series forecasting by integrating temporal, visual, and textual modalities, it has some limitations.

First, the framework performs less robustly on datasets with highly volatile or irregular patterns, such as those with sudden changes or non-stationary trends, compared to datasets with periodic structures. This limitation may arise from the current visual transformation techniques, which may not adequately capture abrupt temporal dynamics or sudden shifts. Future work could refine these transformations to better handle such irregularities.

Second, the current implementation relies on pre-trained VLMs like ViLT and CLIP, which are optimized for natural vision-language tasks rather than time series forecasting. While these models excel in visual understanding, their textual capabilities are limited, often supporting only shorter text inputs and lacking domain-specific knowledge relevant to time series. This restricts their ability to fully utilize textual context for forecasting. Future work could involve developing larger, domain-specific VLMs trained on multimodal time series datasets to address these limitations.

### D.2. Future Work

Building on the current framework, several promising directions for future research emerge:

- **Optimizing Visual Transformations:** Future work could focus on developing adaptive visual transformation techniques that better preserve temporal dynamics, especially for datasets with irregular or non-stationary patterns, to more effectively highlight sudden changes and complex trends.

- **Scaling Multimodal VLMs for Enhanced Forecasting:** While the current framework uses smaller pre-trained Vision-Language Models (VLMs), scaling to larger models could improve forecasting accuracy. Investigating trade-offs between model size, computational efficiency, and performance is a promising direction for future research. Additionally, studying different VLM architectures could identify optimal designs for temporal modeling.

- **Interpretable Multimodal Learning for Time Series Analysis:** Understanding the contributions of visual and textual modalities in time series forecasting is crucial for improving model transparency. Future work could explore the

interpretability of multimodal features, analyzing how different types of information contribute to performance gains. This would provide deeper insights into temporal dependencies and enhance trust in multimodal forecasting models.

- **Pre-training Multimodal Foundation Models for Time Series Analysis:** Existing VLMs are not designed to handle time series data, limiting their ability to capture domain-specific temporal context. Future research could focus on constructing large-scale multimodal datasets that pair time series data with rich textual and visual annotations, enabling the development of models specifically optimized for time series forecasting. Additionally, this multimodal framework could be extended to support multi-task learning, enhancing the model's versatility for tasks such as anomaly detection, classification, or imputation. This would allow the model to capture a broader range of temporal patterns and dependencies, improving its applicability across various domains.

By addressing these directions, future research can build on the foundation laid by Time-VLM, advancing the field of multimodal time series forecasting while ensuring responsible and ethical deployment in real-world applications.

