# OpenReview forum: "Time-VLM: Exploring Multimodal Vision-Language Models for Augmented Time Series Forecasting"
_ICML.cc/2025/Conference — ICML 2025 poster_

### Official Review · Reviewer_wRhk · 2025-03-09

**Overall Recommendation:** 3

**Summary:**

This paper proposes Time-VLM, a multimodal framework using pre-trained VLMs for time series forecasting. It introduces VAL, RAL, and TAL modules to consider the information from three different views. Experimental results show that it achieves good performance with high efficiency. It contributes to a possible new direction in multimodal time series modeling.

## Update After Rebuttal

I've read the rebuttal and other reviewers' comments, and my final rating is weak accept.

**Claims And Evidence:**

The claim that fusing temporal, visual, and textual modalities improves forecasting accuracy is supported by ablation studies. The claims are almost correct. I don’t identify crucial technical errors within the paper.

**Essential References Not Discussed:**

To the best of my knowledge, there are a series of existing studies discussing multimodal time series analysis [1, 2]. It would be good to discuss several of them for a more thorough comparison.

[1] Y. Jiang et al. Multi-modal Time Series Analysis: A Tutorial and Survey. arXiv.

[2] H. Liu et al. How Can Time Series Analysis Benefit From Multiple Modalities? A Survey and Outlook. arXiv.

**Experimental Designs Or Analyses:**

Overall, the experiments are thorough and well-structured, addressing key aspects of time series forecasting and multimodal learning. While the results are promising and the approach demonstrates strong potential, further exploration in more datasets and diverse real-world scenarios could provide additional insights. Nonetheless, the evaluation is solid and offers a convincing case for the effectiveness of the proposed framework.

**Methods And Evaluation Criteria:**

The methods of Time-VLM are primarily divided into the following components:
- VAL’s Image Encoding: The frequency/periodicity-based time-series-to-image conversion aligns well with VLM requirements while retaining temporal relationships.
- TAL’s Hybrid Prompts: Combining statistical features with domain-specific knowledge provides a pragmatic design for real-world deployment.
- RAL’s Memory-Enhanced Modeling: The retrieval-augmented mechanism leverages historical patterns to enhance temporal understanding, offering a robust foundation for forecasting.
These three components are fused via the vision-language model to produce the predictive output. The evaluation metrics used are standard in this field (MSE, MAE, SMAPE, MASE, OWA), and the datasets (ETT, Weather, Electricity, Traffic, M4) are comprehensive and widely recognized.

**Other Comments Or Suggestions:**

N/A

**Other Strengths And Weaknesses:**

Overall, the insight of fusing textual and visual information with time series is interesting. The Time-VLM model can effectively address the complicated relationships among these three modalities. The experiments can verify the efficacy of these modalities.

Despite the above merits, the paper can be improved in the following aspect:
1. The projection from the text/images to the time series space needs more interpretation. Can the paper provide some case studies for a better illustration?
2. According to Table 1 to Table 3, the proposed model can achieve good performance in zero/few-shot settings. However, the results in Table 4 and 5 are not significant, i.e., only limited improvements on the full-shot setting. Can the author clarify this point? In other words, if we have enough data, we don’t need other modalities anymore?
3. How about the efficiency of the proposed model? Do the authors evaluate different model size of Time-VLM on these datasets? How is the sensitivity to the parameter size? Generally the model size should largely impact the model performance.

**Questions For Authors:**

Please make rebuttal to the weaknesses. I’ve listed several questions there. Thank you.

**Relation To Broader Scientific Literature:**

This work aligns with recent efforts to integrate foundation models into time series analysis (e.g., Time-LLM, LLMTime) but stands out by uniquely addressing multimodal fusion. It advances the field by demonstrating how VLMs can be effectively adapted for time series forecasting, bridging gaps between modalities.

**Theoretical Claims:**

N/A

---

> ### Author Rebuttal · Authors · 2025-03-31
>
> > **Q1: Add the latest references related to multi-modal time series**
> >
>
> **A1:** Thank you for your reminder. Our manuscript now includes two recent comprehensive surveys on multi-modal time series analysis [1,2], which we have integrated into both the **Introduction** and **Related Work** sections. These references help contextualize our work within current research trends and highlight important methodological connections.
>
> [1] Y. Jiang et al. Multi-modal Time Series Analysis: A Tutorial and Survey.
>
> [2] H. Liu et al. How Can Time Series Analysis Benefit From Multiple Modalities? A Survey and Outlook.
>
> > **Q2: The projection from the text/images to the time series space needs more interpretation. Can the paper provide some case studies for a better illustration?**
> >
>
> **A2**: Thank you for mention this important question. We have discussed a lot about this point, you can refer to **Q3/A3 (Reviewers 6Pe4 / gzkF)**, hope to address your concern.
>
> > **Q3: According to Table 1 to Table 3, the proposed model can achieve good performance in zero/few-shot settings. However, the results in Table 4 and 5 are not significant, i.e., only limited improvements on the full-shot setting. Can the author clarify this point? In other words, if we have enough data, we don’t need other modalities anymore?**
> >
>
> **A3:** Thank you for your question. The experimental results indeed support this observation. However, it is worth noting that Time-VLM still **outperforms** Time-LLM despite having significantly fewer parameters. Our conclusion is that **when time series data is limited**, visual and textual transformations can **compensate for the lack of information**, though time series data remains the most critical factor. When the time series data is **sufficiently diverse and abundant**, the model can learn **effectively from the time series alone**.
>
> Currently, Time-VLM research is **constrained** by the lack of high-quality multi-modal time series datasets. Existing text and image data are generated, with real-world multi-modal datasets—such as medical scenarios combining ECG time series, textual diagnoses, and other modalities—the fusion of these data types could yield better performance even in full-shot settings, as different modalities would **complement each other** more effectively.
>
> > **Q4: How about the efficiency of the proposed model? Do the authors evaluate different model size of Time-VLM on these datasets? How is the sensitivity to the parameter size? Generally the model size should largely impact the model performance.**
> >
>
> **A4:** The model size of Time-VLM is determined by its **underlying VLM architecture**. We evaluated four variants: ViLT-, CLIP-, BLIP-2-based implementations, and a custom model.
>
> Experiments show that larger models do not perform better. Since time series data is **relatively sparse** compared to multimodal image-text data, **excessively large VLMs** are prone to **overfitting** and **slower efficiency**. Among the tested versions, the ViLT- and CLIP-based configurations achieve the best trade-off between efficiency and performance.

---

### Official Review · Reviewer_Cr31 · 2025-03-10

**Overall Recommendation:** 3

**Summary:**

Time-VLM is a groundbreaking framework that leverages pre-trained VLMs to unify temporal, visual, and textual data for time series forecasting. Key innovations include adaptive time-series-to-image conversion (VAL), memory-enhanced temporal modeling (RAL), and contextual prompt generation (TAL). The experiments demonstrate superior performance in few-shot and zero-shot settings, with efficiency gains compared to Time-LLM. This work opens a new paradigm for multimodal time series forecasting.

**Claims And Evidence:**

The claims are compelling. Ablation studies (Table 6) strongly support the multimodal synergy claim, as removing any modality degrades forecasting accuracy. Time-VLM's efficiency is notable, reducing parameters 20× compared to Time-LLM (Table 7) yet maintaining competitive performance, with inference speed metrics further affirming it. Results on zero-shot cross-domain tasks (Table 3) validate its generalization ability. Overall, the evidence substantiates the approach's effectiveness.

**Essential References Not Discussed:**

1) Vision Language Models Are Few-Shot Audio Spectrogram Classifiers (NeurIPS Workshop 2024):
This work evaluates zero-shot capabilities of LMMs for audio classification using spectrogram images and textual prompts. While related, Time-VLM uniquely addresses forecasting-specific challenges (e.g., periodicity-aware image encoding in VAL), highlighting its broader applicability.

2) Training-Free Time-Series Anomaly Detection: Leveraging Image Foundation Models (Arxiv 2024):
This work focuses on zero-shot anomaly detection using visual representations. Discussing it would further emphasize Time-VLM’s unique contributions in multimodal forecasting and its efficiency compared to larger LMMs.

**Experimental Designs Or Analyses:**

The experiments are comprehensive and well-executed, spanning key aspects such as long/short-term forecasting, few/zero-shot prediction, ablation studies, model analysis, visualization case studies, and hyperparameter sensitivity.

**Methods And Evaluation Criteria:**

The methods are technically sound. VAL's time-series-to-image conversion retains temporal relationships, suiting VLMs. TAL's hybrid prompts combine statistical and domain knowledge practically. RAL's memory-enhanced modeling uses historical patterns for forecasting. Multimodal fusion of RAL, VAL, and TAL modalities captures diverse patterns, improving accuracy. Evaluation is comprehensive, with standard metrics and diverse datasets.

**Other Comments Or Suggestions:**

Please refer to the weakness.

**Other Strengths And Weaknesses:**

Strengths:

1) Novel use of VLMs for time series, enabling semantic context injection.

2) Strong empirical validation across diverse domains.

Weaknesses:

1) Scalability: Exploring larger VLMs (e.g., LLaVA, GPT-4V) could enrich textual context and further improve performance, offering a promising avenue for future work.

**Questions For Authors:**

Please refer to the weakness.

**Relation To Broader Scientific Literature:**

This work aligns with recent efforts to integrate foundation models into time series analysis (e.g., Time-LLM, LLMTime) but stands out by uniquely addressing multimodal fusion. It advances the field by demonstrating how VLMs can effectively adapt to time series forecasting, bridging gaps between modalities.

**Theoretical Claims:**

No theoretical claims.

---

> ### Author Rebuttal · Authors · 2025-03-31
>
> > **Q1: Scalability: Exploring larger VLMs (e.g., LLaVA, GPT-4V) could enrich textual context and further improve performance, offering a promising avenue for future work.**
> >
>
> **A1:** Thank you for your suggestion. We empirically evaluated VLMs across different scales, ranging from smaller architectures like ViLT (143M) to larger ones such as BLIP-2 (3.75B). Our experiments revealed two key findings:
>
> 1. **Diminishing Returns on Scale**: Larger models demand substantially more computational resources without delivering performance gains.
> 2. **Overfitting Tendency**: On benchmarks like the ETT datasets, we observed rapid training loss reduction alongside increasing validation loss, indicating overfitting.
>
> Based on these results, we adopted a more compact VLM to optimize the trade-off between efficiency and effectiveness. We hypothesize that the rich **multimodal priors from large-scale pretraining** may be unnecessarily **complex** for our current datasets (ETT, Traffic, Weather, Electricity), which are **relatively small-scale**.

---

### Official Review · Reviewer_gzkF · 2025-03-10

**Overall Recommendation:** 3

**Summary:**

This paper proposes Time-VLM, a multimodal framework that leverages vision-language models to encode temporal, "visual" and textual modalities for enhanced time series forecasting. Specifically, RAL encodes and saves time series into a memory bank for further interaction with multi-modal embedding. VAL encodes time series to multi-dimensional representations and TAL generates time series descriptions. "vision" and textual features are processed with frozen pretrained VLMs to obtain informed meta representations. Fused with temporal features, the Time-VLM achieves superior performance on various experiment settings.

**Claims And Evidence:**

Yes. Claims are clear and well supported.

**Essential References Not Discussed:**

N/A

**Experimental Designs Or Analyses:**

Yes, experients are sound and comprehensive.

**Methods And Evaluation Criteria:**

Yes, methods and evaluation criteria make sense in this area.

**Other Comments Or Suggestions:**

N/A

**Other Strengths And Weaknesses:**

Strength:

1. The n-dimensional representation of time series is good. Instead of merely plotting the time series, it allows for a more effective utilization of the vision encoder.
2. Comprehensive and complete experiments.
3. The framework is new to this field.

Weakness:

1. The textual encoder component follows Time-LLM, and based on your ablation study, its contribution appears marginal. There is room for improvement in this area.
2. The n-dimensional representation of time series and its description are loosely connected. Directly utilizing a pretrained VLM may reduce effectiveness.
3. The “visual” representation of time series is difficult for human perception to comprehend. The visualization in C.1 lacks meaningful interpretation.

**Questions For Authors:**

1. Would using a pretrained VM combined with a pretrained LM, instead of a pretrained VLM, better enhance your visual and textual inputs?
2. Since you mentioned foundation models for time series, could they offer stronger encoding capabilities and enhance your pipeline?

**Relation To Broader Scientific Literature:**

Contribute to multi-modal time series analysis via combining temporal, visual and textual modalities

**Theoretical Claims:**

N/A

---

> ### Author Rebuttal · Authors · 2025-03-31
>
> Thank you for recognizing the method design, complete evaluation and unique innovation of our paper. Below are our responses to your major questions.
>
> > **Q1: Textual Encoder Limitations Analysis**
> >
>
> **A1:** We systematically investigated the limitations of textual encoders in VLMs through three complementary analyses:
>
> 1. **Input Length Analysis**: We evaluated three VLMs with progressively longer maximum text input lengths—**ViLT (40 tokens), CLIP (77 tokens), and BLIP-2 (512 tokens)**—and observed no improvement in forecasting performance despite increased model capacity. Instead, larger models exhibited lower efficiency, with significantly slower inference and training speeds.
> 2. **Custom VLM Analysis**: We implemented a Custom implementation on single VM (ViT-B/16) + LM (BERT-Base) on traffic datasets. However, experiments revealed that it underperformed native VLMs.
>
>
>     | Horizon | Time-VLM (ViLT) | Time-VLM (Custom) |
>     | --- | --- | --- |
>     | 96 | 0.148 | 0.158 |
>     | 192 | 0.193 | 0.206 |
>     | 336 | 0.243 | 0.258 |
>     | 720 | 0.312 | 0.334 |
>     | Avg | 0.224 | 0.239 |
> 3. **Embedding Space Divergence**: As we discussed in **Q3/A3 of Reviewer 6Pe4**, The t-SNE visualization demonstrates that:
>     - **Text embeddings (blue)** from COCO-Text form distinct clusters separated from time-series data distributions (ETT/Traffic/ECL), confirming poor semantic alignment.
>     - **Visual embeddings (green)** from COCO-Image naturally encompass time-series features, validating that pixel-space representations better preserve temporal dynamics.
>
>     This reveals a lack of effective cross-modal alignment. Crucially, the inherent divergence between image, text, and time-series modalities—compounded by their distinct training data distributions—hinders performance without explicit alignment mechanisms.
>
> 4. **Future Potential**: Time-VLM is a **self-enhanced** paradigm that operates without real image/text data—both are generated from raw time series. We believe that with a high-quality dataset covering real-world multimodal time-series data (e.g., ECG time series paired with medical images and diagnostic reports), Time-VLM’s potential could be fully unlocked. At that stage, modalities could complement each other more effectively.
>
> > **Q2: The n-dimensional representation of time series and its description are loosely connected. Directly utilizing a pretrained VLM may reduce effectiveness.**
> >
>
> **A2:** Your concern is right. However, our approach does not aim to map time series directly to text. Instead, we project them into a joint text-image multi-modal space, leveraging the complementary strengths of both modalities. For a deeper discussion on this methodology, please refer to **Q3/A3 (Reviewer 6Pe4)**.
>
> > **Q3: The “visual” representation of time series is difficult for human perception to comprehend. The visualization in C.1 lacks meaningful interpretation.**
> >
>
> **A3:** I fully understand your concerns. In response, we added an insightful analysis in **Q3/A3 (Reviewer 6Pe4)**, showing that the pre-training knowledge from the VLM **aligns** with the time series domain. Additionally, our Time-VLM's image generation framework integrates innovative techniques to better align the distribution of the VLM’s input **while maximizing** temporal information retention:
>
> 1. **Frequency Domain Information Injection:**
>     - High-frequency components **appear as** fine-grained textures
>     - Low-frequency components **form** broad color gamut distributions
> 2. **Multi-Scale Period Encoding**:
>     - Cosine/sine functions encode periodic patterns (e.g., daily/weekly cycles)
>     - **Represented** through distinct spatial patterns in the image
> 3. **Interpolation Alignment Mechanism**:
>     - Bilinear interpolation ensures smooth pixel transitions
>     - Sudden time-series changes **correspond to** sharp color intensity variations
> 4. **Color Semantic Mapping**:
>     - Dark blue → low values; light yellow → high values
>     - Gradient transitions **directly indicate** trend directions
>
> We acknowledge that visual interpretability remains a challenge and plan to explore this further in future work.
>
> > **Q4: Would using a pretrained VM combined with a pretrained LM, instead of a pretrained VLM, better enhance your visual and textual inputs?**
> >
>
> **A4:** This is **an interesting idea**, and we **have explored it experimentally**. You can find details in **Q1/A1** and **Q3/A3 (Reviewer 6Pe4)**.
>
> > **Q5: Since you mentioned foundation models for time series, could they offer stronger encoding capabilities and enhance your pipeline?**
> >
>
> **A5:** As discussed in **Q1/A1 (Reviewer 6Pe4)**, Time-VLM and time-series foundation models follow **different paradigms**. However, **we may explore building a pre-trained vision-language-enhanced foundation model** once high-quality multimodal time-series datasets become available.

---

> > ### Comment · Reviewer_gzkF · 2025-04-07
> >
> > Thank you to the author for the response. It addressed most of my concerns. I will keep my scores.

---

### Official Review · Reviewer_6Pe4 · 2025-03-14

**Overall Recommendation:** 2

**Summary:**

The paper introduces Time-VLM, a multimodal time series forecasting framework that leverages pre-trained Vision-Language Models (VLMs) to integrate temporal, visual, and textual information. By combining retrieval-augmented learning, vision-based encoding, and text-based contextualization, Time-VLM enhances forecasting performance, particularly in few-shot and zero-shot settings.

**Claims And Evidence:**

Overall, the claims made in the paper are clear.

**Essential References Not Discussed:**

The paper would benefit from citing and discussing prior works on zero-shot and few-shot forecasting, particularly those related to foundation models in time series forecasting. Several recent studies, such as CHRONOS, TimesFM, and MOIRAI, have explored the use of large-scale pre-trained models for improved generalization in forecasting tasks.

**Experimental Designs Or Analyses:**

I have reviewed the experimental design and analyses. To further establish the validity of the results, additional comparisons with other models that perform well in zero-shot and few-shot forecasting would be necessary.

**Methods And Evaluation Criteria:**

Yes, the proposed methods and evaluation criteria are appropriate for the problem of time series forecasting.

**Other Comments Or Suggestions:**

N/A

**Other Strengths And Weaknesses:**

Strengths
1. The paper presents a clear motivation and is well-structured, making it easy to follow.
2. The approach of transforming time series into visual cues and leveraging a memory bank for enriched feature extraction is novel.
3. The proposed method demonstrates strong performance across various time series forecasting scenarios, highlighting its effectiveness.

Weaknesses
1. Comparison with Time Series Foundation Models: Recent foundation models for time series forecasting (e.g., CHRONOS, TimesFM, MOIRAI) have demonstrated strong zero/few-shot forecasting performance. A direct comparison with these models is necessary to validate the effectiveness of the proposed approach.

2. Instance Normalization and Misalignment: Since raw time series data undergoes normalization, but textual information is extracted from the unnormalized version, this may cause misalignment in feature representations. The paper should include experiments addressing this issue to ensure robustness.

3. Role of the Vision Encoder: It is unclear whether a pre-trained vision encoder can effectively process images generated from time series data. A more thorough evaluation is needed to justify its role. Additionally, the decision to freeze the vision encoder should be explained, and results from fine-tuning should be provided for comparison.

4. Lack of Component Analysis: While the paper provides extensive forecasting results, there is a lack of detailed ablation studies on key components. For example, the memory bank is a core part of the method, yet its contribution is not analyzed enough. Further component-wise evaluations would enhance the credibility of the approach.

**Questions For Authors:**

Please refer to the Weaknesses.

**Relation To Broader Scientific Literature:**

This paper contributes to the broader scientific literature by exploring multimodal time series forecasting with pre-trained Vision-Language Models (VLMs). While previous works have integrated either text or vision with time series forecasting, this study is one of the first to leverage both modalities simultaneously.

**Theoretical Claims:**

The paper does not contain any theoretical claims that require formal proof verification.

---

> ### Author Rebuttal · Authors · 2025-03-31
>
> Thank you for recognizing the clear motivation, method design, and rigorous evaluation of our paper. Below we endeavor to address your questions.
>
> > **Q1: Comparison with Time Series Foundation Models**
> >
>
> **A1:** We appreciate your suggestion. However, ​Time-VLM fundamentally **differs** from foundation models (e.g., CHRONOS, TimesFM, MOIRAI) in its **​learning paradigm**: While the latter rely on ​**external datasets** for pre-training, Time-VLM ​uses only the target dataset. By generating text and visual modalities, it augments raw time series ​**without external knowledge**, enabling efficient fine-tuning and cross-dataset zero-shot learning.
>
> However, we added a comparison of Time-VLM (5% few-shot) vs. foundation models (zero-shot) on ETT datasets. Results show Time-VLM achieves lower MSE across all datasets, validating its effectiveness under data scarcity.
>
> | Methods | Time-VLM | Moirai | Chronos | TimesFM |
> | --- | --- | --- | --- | --- |
> | ETTh1 | 0.442 | 0.475 | 0.560 | 0.489 |
> | ETTh2 | 0.354 | 0.379 | 0.392 | 0.396 |
> | ETTm1 | 0.364 | 0.714 | 0.636 | 0.434 |
> | ETTm2 | 0.262 | 0.343 | 0.313 | 0.320 |
>
> > **Q2: Instance Normalization and Misalignment**
> >
>
> **A2:** We added ablation experiments on Weather dataset to evaluate text normalization’s impact. Results show that normalized text **reduces MAE by 1.29%**, underscoring the importance of cross-modal alignment. We will fix this in final version.
>
> | Horizon | Time-VLM (Raw) | Time-VLM (Normalized) |
> | --- | --- | --- |
> | **96** | 0.160 | 0.159 |
> | **192** | 0.203 | 0.201 |
> | **336** | 0.253 | 0.249 |
> | **720** | 0.317 | 0.312 |
> | Avg | 0.233 | 0.230 |
>
> > **Q3: Role of the Vision Encoder and Freezing Rationale**
> >
>
> **A3:** We conducted an analysis to evaluate how pre-trained VLMs can be applied to time series forecasting. Specifically, we sampled 200 **image-text pairs from MSCOCO** (VLM’s pre-training dataset) and 60 samples each from **time series dataset (ETT, Traffic, Weather, ECL)**. Using t-SNE, we visualized four embedding types in 2D space:
>
> 1. Multi-modal embeddings from COCO-Pair samples through VLM ⇒ representing **VLM’s pre-training knowledge**
> 2. Multi-modal embeddings from time series-generated image-text pairs through VLM ⇒ representing **Time-VLM’s augmented knowledge**
> 3. Visual embeddings from COCO-Image samples through ViT ⇒ representing **visual knowledge from a single VM**
> 4. Text embeddings from COCO-Text samples through BERT ⇒ representing **text knowledge from a single LM**
>
> Key findings from the visualization https://anonymous.4open.science/r/Time-VLM/ts_embeddings_with_coco_samples.png:
>
> 1. **COCO-Image features (green)** fcluster centrally, **surrounded by time-series features** (ETT: yellow/orange, ECL: purple, etc.), confirming intrinsic visual-temporal similarity.
>
>     This aligns with VisionTS [1], where **visual features** (pixel intensity variations, repeating patterns, color consistency, and edge transitions) **directly map** to **temporal behaviors** (value fluctuations, periodicity, stable segments, and outliers/abrupt changes, respectively). This explains why time series imaging [2] has emerged as a hot research direction. Our image generation process further advances this explicitly encoding frequency/periodicity and using interpolation/color semantics to preserve temporal information.
>
> 2. **COCO-Text features (blue)** show **clear separation** with time-series clusters, highlighting modality gaps.
> 3. **COCO-Pair features (red)** achieve **maximal overlap** with time-series data, demonstrating **cross-modal complementarity** - text semantics enhance visual-temporal alignment.
>
> This **motivates** Time-VLM. Compared to Time-LLM and VisionTS which utilize a single modality projection, we argue that **projecting time series into multi-model embeddings** is more plausible. Time-VLM's remarkable zero-shot and few-shot learning capabilities primarily stem from the pre-trained VLM's knowledge.
>
> **Regarding freezing the visual encoder**, we did extensive experiments, which revealed that unfreezing led to training instability (e.g., overfitting on small datasets like ETT). Since the VLM has already achieved cross-modal alignment, freezing the encoder allows us to reuse this capability while avoiding excessive training overhead, we only add a simple fusion layer that is time-efficient for alignment.
>
> [1] M. Chen et al. VisionTS: Visual Masked Autoencoders Are Free-Lunch Zero-Shot Time Series Forecasters
>
> [2] J. Ni et al. Harnessing Vision Models for Time Series Analysis: A Survey.
>
> > **Q4: Lack of Component Analysis**
> >
>
> **A4:** We expanded ablation studies on the memory bank. Results on the Weather dataset (MSE) confirm **both local and global memory are critical**:
>
> | Horizon | Full Model | w/o Local Mem | w/o Global Mem |
> | --- | --- | --- | --- |
> | 96 | 0.160 | 0.185 | 0.165 |
> | 192 | 0.203 | 0.235 | 0.210 |
> | 336 | 0.253 | 0.295 | 0.265 |
> | 720 | 0.317 | 0.375 | 0.330 |
> | Avg | 0.233 | 0.273 | 0.243 |

---

### Decision · Program_Chairs · 2025-05-01

**Decision:**

Accept (poster)

**Comment:**

This paper proposes a multimodal framework using pre-trained vision-language models (VLMs) for time series forecasting.

After the rebuttal, three reviewers (Cr31, gzkF, and wRhk) gave positive ratings (3, 3, 3), recognizing the paper’s strong performance and novel ideas. Reviewer 6Pe4 also acknowledged the novelty and good performance of the proposed method but raised concerns about the lack of comparison with foundation models and the absence of certain analyses. The authors provided a thorough response; however, Reviewer 6Pe4 did not reply to the feedback.

The Area Chair (AC) agrees with most of the reviewers and recommends accepting the paper. Additionally, it is strongly suggested that the authors incorporate the content discussed in the feedback into the revised version to make the paper more comprehensive.